# DCAF14 regulates CDT2 to promote SET8-dependent replication fork protection

Neysha Tirado-Class, Caitlin Hathaway, Anthony Nelligan, Thuan Nguyen, Huzefa Dungrawala

DDB1- and CUL4-associated factors (DCAFs) CDT2 and DCAF14 are substrate receptors for Cullin4–RING E3 ubiquitin ligase (CRL4) complexes. CDT2 is responsible for PCNA-coupled proteolysis of substrates CDT1, p21, and SET8 during S-phase of cell cycle. DCAF14 functions at stalled replication forks to promote genome stability, but the mechanism is unknown. We find that DCAF14 mediates replication fork protection by regulating CRL4$^{CDT2}$ activity. Absence of DCAF14 causes increased proteasomal degradation of CDT2 substrates. When forks are challenged with replication stress, increased CDT2 function causes stalled fork collapse and impairs fork recovery in DCAF14-deficient conditions. We further show that stalled fork protection is dependent on CDT2 substrate SET8 and does not involve p21 and CDT1. Like DCAF14, SET8 blocks nuclease-mediated digestion of nascent DNA at remodeled replication forks. Thus, unregulated CDT2-mediated turnover of SET8 triggers nascent strand degradation when DCAF14 is absent. We propose that DCAF14 controls CDT2 activity at stalled replication forks to facilitate SET8 function in safeguarding genomic integrity.

## Introduction

Faithful DNA duplication is critical for genome stability and cellular fitness. During DNA synthesis, several types of chromatin lesions cause replication stress by hindering progression of replisomes. To preserve stability of replication machinery and prevent mistakes in propagating genetic and epigenetic information, cells employ a wide variety of mechanisms to sense the damage event, repair the lesion, and resume DNA synthesis (Zeman & Cimprich, 2014; Cortez, 2019). Central to these mechanisms are stress response proteins that are actively engaged to stabilize stalled replication forks and regulate protein function by posttranslational modifications such as phosphorylation and ubiquitination. Protein ubiquitination, catalyzed by E3 ubiquitin ligases, modulates replication dynamics at both unperturbed and damaged forks to control genome stability (Mirsanaye et al, 2021). Although canonical ubiquitin signaling during DNA replication and replication-linked processes is

associated with proteolysis, degradation-independent ubiquitin signaling plays equally critical roles in fork protective mechanisms such as translesion synthesis, Fanconi anemia pathway, and replication fork reversal.

CRL4$^{CDT2}$, a CUL4–DDB1 E3 ubiquitin ligase complex, functions in genome maintenance primarily by catalyzing proteolytic degradation of substrates via polyubiquitination (Abbas et al, 2008; Havens & Walter, 2011). During S-phase, CRL4$^{CDT2}$ triggers destruction of substrates that harbor PCNA-interacting domains called PIP degrons (Havens & Walter, 2009). Unlike a canonical PIP box consensus sequence, PIP degrons consist of a PIP box with a basic arginine/lysine residue four amino acids downstream of the PIP box (Havens & Walter, 2009). Substrate ubiquitination by CRL4$^{CDT2}$ strictly occurs on substrates interacting with chromatin-bound PCNA (Arias & Walter, 2006; Abbas et al, 2008). Some of the well-characterized substrates include histone methyltransferase SET8 (Abbas et al, 2010; Centore et al, 2010), origin-licensing factor CDT1 (Arias & Walter, 2006; Guarino et al, 2011), and CDK inhibitor p21 (Abbas et al, 2008; Kim et al, 2008). By regulating the stability of these proteins, CRL4$^{CDT2}$ imposes DNA replication to occur only once per cell cycle and controls cell cycle progression (Abbas & Dutta, 2011). Thus, mis-regulation in CDT2 activity results in genome instability and is also linked to tumorigenesis (Abbas & Dutta, 2011; Panagopoulos et al, 2020).

CRL4$^{CDT2}$-driven proteolytic degradation occurs even in DNA damage conditions. Upon exposure to genotoxic stress, a similar mechanism is used wherein chromatin bound-PCNA serves as a platform for the CRL4$^{CDT2}$ complex to target SET8 (Centore et al, 2010; Jorgensen et al, 2011), CDT1 (Higa et al, 2006a; Hu & Xiong, 2006; Ralph et al, 2006) and p21 (Nishitani et al, 2008). Despite being subjected to proteasomal turnover, studies indicate that docking onto PCNA provides CDT2 substrates with an opportunity to regulate repair on DNA damage sites. For instance, monomethylation of histone H4 on lysine 20 (H4K20) by SET8 promotes 53BP1 recruitment to double-strand breaks (DSBs) (Oda et al, 2010). p21 regulates histone acetyltransferase activity of p300 at UV damage sites (Perucca et al, 2006; Cazzalini et al, 2008) and its interaction with PCNA modulates translesion synthesis to control mutagenic load (Avkin et al, 2006). Similar to p21, CDT1 also influences translesion synthesis but the contributions of CDT1 proteolysis by CDT2 remain unclear. In human cells, the kinetics of CDT1 degradation is

Department of Molecular Biosciences, University of South Florida, Tampa, FL, USA

Correspondence: hdungrawala@usf.edu

up-regulated in response to UV damage (Tsanov et al, 2014). The rapid clearance of PIP-degron substrates facilitates focal recruitment of PIP-box containing translesion polymerases to PCNA because PIP-degrons display 10-fold higher affinity for PCNA compared with canonical PIP boxes (Havens & Walter, 2009; Michishita et al, 2011). In contrast, CDT1 degradation is subdued in fission yeast upon UV exposure (Guarino et al, 2011). The aforementioned observations imply that the destructive function of CRL4$^{CDT2}$ must be delicately controlled such that the substrates, before being degraded, can exert their function locally at DNA damage sites to facilitate certain repair processes.

DCAF14 is a substrate receptor for CRL4 complexes and is one of several DCAFs, which also includes CDT2 (known as DCAF2 or DTL) (Higa et al, 2006b; Jin et al, 2006). Mechanistically, DCAFs partner with the specific substrates and assemble on CUL4 scaffold via the adaptor proteins DDB1 to form a CRL4 complex that engages in ubiquitination of the acquired substrates. Recently, DCAF14 was identified as a replication stress response protein that is recruited to stalled replication forks and protects newly synthesized DNA from nuclease-mediated degradation (Townsend et al, 2021). Depletion of CRL4 components CUL4B and DDB1 also trigger nascent DNA digestion suggesting that DCAF14 mediates replication fork protection in a CRL4-dependent manner. Importantly, DCAF14 promotes stalled fork stability because DCAF14 inactivation causes DSBs resulting in genomic instability. However, a functional link between DCAF14 and replication fork protection remains to be identified.

In this study, we report that fork protection defects in DCAF14-deficient cells occur because of aberrant increase in CRL4$^{CDT2}$ activity. We show that loss of DCAF14 results in increased turnover of CDT2 substrates SET8, p21, and CDT1. Blocking CRL4$^{CDT2}$ activity using MLN4924 or by depleting CDT2 reverses the exacerbated loss of CDT2 substrates in cells devoid of DCAF14. Whereas replication elongation remains unperturbed in DCAF14-depleted cells, the heightened CDT2 function triggers nascent strand degradation and stalled fork instability in DCAF14-deficient conditions. Importantly, in the absence of CDT2, restoration of fork protection in DCAF14-depleted cells is mediated by CDT2 substrate SET8. These studies uncover the need to fine-tune CRL4$^{CDT2}$ activity by DCAF14 at stalled replication forks and establish SET8 as a mediator of nascent DNA stability.

# Results

## Loss of DCAF14 diminishes monomethylation of H4K20

CRL4 function is linked to histone methylation in cells (Higa et al, 2006b). Several DCAFs consist of WD40-repeat domains that alter histone methylation states on lysine residues. Specifically, WDR5 and RBBP5 of the mixed-lineage leukemia histone methylation complex catalyze histone H3 methylation on lysine K4 (H3K4) that occupies transcription start sites (Wysocka et al, 2005) and hydroxyurea (HU)-stalled replication forks (Ray Chaudhuri et al, 2016). Consistent with previous studies, reduction in H3K4me3 is specific to depletion of WDR5 and RBBP5 in U2OS cells (Fig 1A). To determine whether substrate receptor DCAF14 has a role in regulating histone

methylation, we compared steady-state levels of multiple mono- and tri-methylation marks on histones H3 and H4 using RIPA-extracted whole-cell lysates from siDCAF14-transfected U2OS cells (Fig 1B). We observe that DCAF14 depletion results in substantial loss of H4K20me1 (Fig 1C). The decrease in H4K20me1 levels is comparable to cells in which function of H4K20 mono-methyltransferase SET8 is down-regulated either by siRNA-mediated depletion or acute enzymatic inhibition using substrate-competitive inhibitor UNC0379 (Ma et al, 2014) (Fig 1C). Immunofluorescence staining and quantitative analyses of detergent-resistant H4K20me1 also show similar results (Fig 1D) and is observable in HeLa cells indicating these effects are not cell type-specific (Fig 1D). In U2OS cells transfected with siRNAs targeting 5′ untranslated region (UTR) of DCAF14, complementation with DCAF14 cDNA partially restores H4K20me1 expression (Fig 1E), indicating that monomethylation of H4K20 is regulated by DCAF14.

## DCAF14 deficiency escalates SET8 turnover

While we were investigating changes in H4K20me1, we observed lower levels of SET8 when DCAF14 was depleted (Fig 1C). SET8 is also down-regulated in whole cell lysates extracted from siDCAF14-transfected U2OS, HeLa, and RPE1 cells (Fig 2A and B). In comparison with SET8, levels of H4K20me1 demethylase PHF8 remain unchanged indicating that lower H4K20me1 levels in DCAF14-deficient cells is not because of elevated PHF8 expression. SET8 and corresponding H4K20me1 levels peak during mitosis for chromatin compaction and are reestablished after each round of DNA replication (Rice et al, 2002; Alabert et al, 2015). Thus, we considered the possibility that cell cycle differences account for the observed decrease in SET8 and H4K20me1 in DCAF14-deficient cells. Analysis of cell cycle distribution profiles by flow cytometry using cells dually stained with propidium iodide (PI) and S-phase marker 5-ethynyl-2′-deoxyuridine (EdU) show modest changes in S-phase distributions for siDCAF14-transfected U2OS cells and DCAF14 KO cells (Fig S1A). These results suggest that alterations in H4K20me1 and SET8 are not largely attributed to cell cycle differences. Because SET8-dependent H4K20 monomethylation occurs on chromatin (Nishioka et al, 2002; Rice et al, 2002), we assessed nuclear intensities of SET8 by immunofluorescence. Similar to our observations for H4K20me1 (Fig 1D), chromatin association of SET8 is reduced in DCAF14 KO cells (Fig 2C). Furthermore, lower SET8 levels are rescued in DCAF14 cDNA-transfected KO cells (Fig 2C) indicating that H4K20me1 down-regulation is a consequence of suppressed SET8 levels in DCAF14-depleted cells.

SET8 is subjected to proteolytic degradation by CRL4$^{CDT2}$ during DNA replication (Abbas et al, 2010; Centore et al, 2010; Jorgensen et al, 2011). To test whether SET8 down-regulation caused by the absence of DCAF14 occurs in a replication-dependent manner, we performed quantitative imaging to analyze SET8 intensities in EdU+ and EdU− nuclei. Immunostaining endogenous SET8 shows consistent patterns with cell cycle distribution, with lower intensities in replicating cells compared to nonreplicating cells (Fig 2D and E). In cells devoid of DCAF14, SET8 expression levels are lower in EdU+ and EdU− nuclei when compared with DCAF14-proficient cells (Fig 2E). In whole-cell lysates, MG132 treatment rescues lower SET8 levels in DCAF14-deficient cells indicating SET8 is subjected to increased

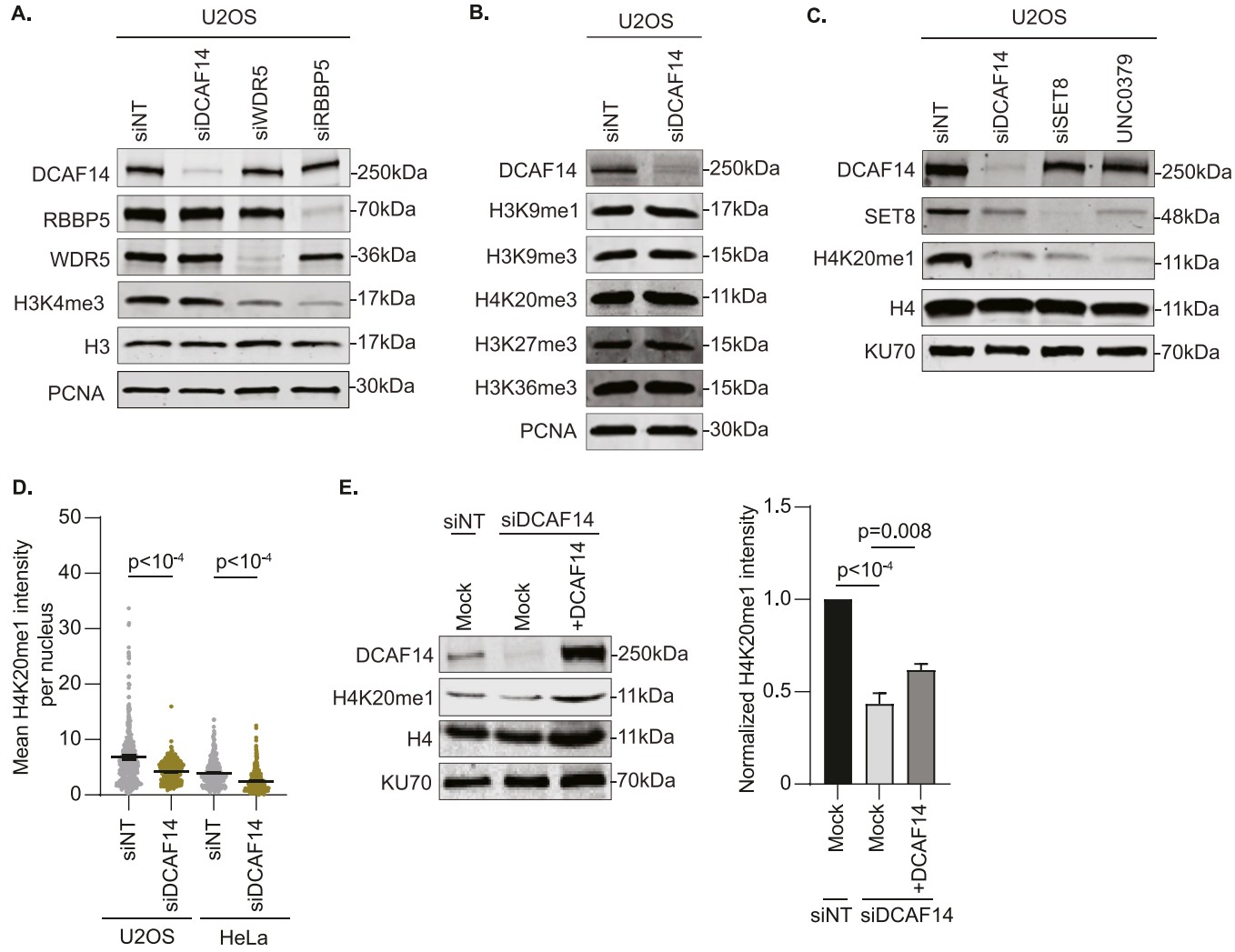

**Figure 1. DCAF14 regulates monomethylation of H4K20.**
**(A)** Whole-cell lysates were extracted from U2OS cells transfected with the indicated siRNAs. Immunoblots were probed with the antibodies as shown. PCNA serves as a loading control. **(B)** Whole-cell lysates were extracted from siNT- and siDCAF14-transfected U2OS cells. Immunoblots were probed with the several histone methylation antibodies as shown. PCNA serves as a loading control. **(C)** U2OS cells were either transfected with the indicated siRNAs or treated with SET8-inhibitor UNC0379 for 4 h. Immunoblots were probed with the antibodies as shown. KU70 serves as a loading control. **(D)** siNT- and siDCAF14-transfected U2OS or HeLa cells were subjected to immunofluorescence analysis. Cells were immunostained for H4K20me1. Mean nuclei intensity was measured by quantitative imaging using DAPI-stained nuclei. Graphs represent mean ± SEM using at least 450 nuclei. **(E)** Whole-cell lysates were extracted from siNT- and siDCAF14 (5'UTR)-transfected U2OS cells that were either mock transfected or overexpressing DCAF14. Immunoblots were probed with the antibodies as shown. KU70 serves as a loading control. Graph represents normalized H4K20me1 intensities to histone H4 from three biological replicates.
Source data are available for this figure.

proteasomal degradation when DCAF14 is lost (Fig S1B and C). Furthermore, rescue of SET8 levels occurs in replicating and nonreplicating populations because transient exposure to either MG132 or cullin inhibitor MLN4924 increases SET8 levels in EdU+ and EdU− nuclei (Figs 2F and S1D). Thus, DCAF14 absence causes increased proteasomal turnover of SET8 during DNA replication and outside of S-phase.

### DCAF14 loss increases CDT2-mediated proteolysis

During DNA replication, CRL4$^{CDT2}$ recognizes PCNA-bound, PIP-degron containing substrates to catalyze their proteolytic destruction by polyubiquitination (Havens & Walter, 2009, 2011). Thus, we investigated the effects of DCAF14 loss on additional CDT2 substrates p21 and CDT1. Whole-cell lysates extracted from DCAF14-deficient cells exhibit lower levels of CDT1 and p21 (Fig 3A and B) and is observable in EdU− and EdU+ nuclei by quantitative imaging (Fig 3C). CDT2 substrates are also decreased in siDCAF14-transfected cells treated with high-dose HU indicating that the substrate depletion remains unaffected in the presence of added replication stress (Fig 3A). To test whether the alterations in p21 and CDT1 in DCAF14 absence occur during DNA replication, we blocked proteolytic turnover transiently using MG132. Like SET8, acute exposure to proteasomal inhibitor MG132 largely restores CDT1 and p21 levels in DCAF14-silenced,

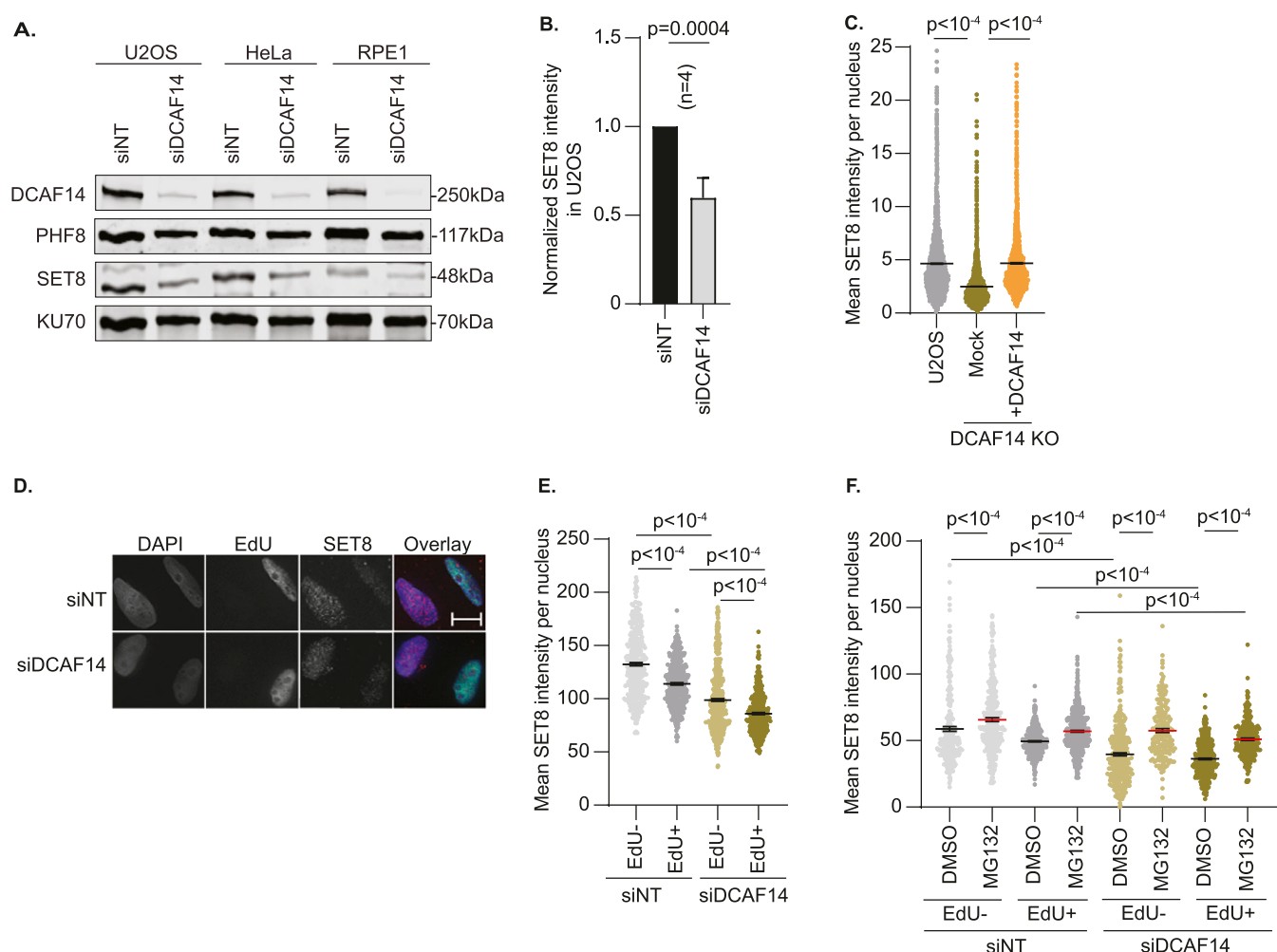

**Figure 2. DCAF14 prevents increased turnover of SET8.**
**(A)** Whole-cell lysates were extracted from U2OS, HeLa or hTERT-RPE1 cells transfected with the indicated siRNAs. Immunoblots were probed with the antibodies as shown. KU70 serves as a loading control. **(B)** Whole cell lysates were extracted from siNT- and siDCAF14-transfected U2OS cells to analyze changes in SET8. Graph represents normalized SET8 intensities from four biological replicates. **(C)** Parental U2OS, DCAF14 KO, and DCAF14 cDNA-transfected KO cells were immunostained for SET8. Mean nuclei intensity was measured by quantitative imaging using DAPI-stained nuclei. Graphs represent mean ± SEM using at least 3,500 nuclei. **(D)** Representative immunofluorescence images of siNT- and siDCAF14-transfected U2OS cells stained for DAPI, EdU, and SET8 are shown with overlay images. Scale bar = 10 $\mu$m. **(E)** siNT- and siDCAF14-transfected U2OS cells were pulsed with EdU for 30 min before immunofluorescence analyses. Mean nuclei intensity of SET8 was measured by quantitative imaging after preselecting EdU+ and EdU− nuclei. Graphs represent mean ± SEM using at least 250 nuclei. **(F)** siNT- and siDCAF14-transfected U2OS cells were pretreated with either DMSO or MG132 for 2 h and pulsed with EdU during the last 30 min of treatment. Cells were immunostained for SET8 and mean nuclei intensity was measured by quantitative imaging after preselecting EdU+ and EdU− nuclei. Graphs represent mean ± SEM using at least 250 nuclei.
Source data are available for this figure.

S-phase cells (Fig 3D). In addition, concomitant exposure to HU and MG132 for 4 h mostly rescues expression of CDT2 substrates in preselected EdU+ nuclei (Fig S2A). Similar protein down-regulation of CDT2 substrates is also observed in DCAF14 KO cells with or without HU treatment (Figs 3E and S2B). These results indicate that the increased turnover of SET8, CDT1, and p21 in DCAF14-deficient conditions is partly linked to aberrant proteolysis by CDT2 in S-phase.

Excessive CDT2-mediated degradation of substrates in DCAF14-deficient cells should thus be ameliorated upon silencing CDT2. To test this prediction, we depleted CDT2 using RNA interference (Fig 3F). Whereas SET8 levels are minimally altered, CDT2-depleted whole-cell extracts exhibit marked accumulation of p21 consistent with activation of G2/M checkpoint in cells devoid of CDT2 (Jin et al, 2006). In addition

to CRL4$^{CDT2}$, SCF-Skp2 targets CDT1 for destruction in S and G2 phases thereby reducing CDT1 levels in CDT2-deficient cells (Nishitani et al, 2006). Significantly, when compared to cells without DCAF14, cells co-depleted of CDT2 and DCAF14 exhibit increased levels of all CDT2 substrates (Fig 3G). Furthermore, this rescue is observable in S-phase by quantitative imaging (Fig 3G) and also occurs in the presence of added replication stress (Fig S2C). These data suggest that CDT2 function is up-regulated in DCAF14-deficient cells and represents a role of DCAF14 in attenuating CDT2 activity. If so, then DCAF14 overexpression should trigger accumulation of CDT2 substrates. Using SET8 as a representative CDT2 substrate, cDNA-mediated overexpression of DCAF14 causes substantial SET8 accumulation in unperturbed conditions and conditions of HU and

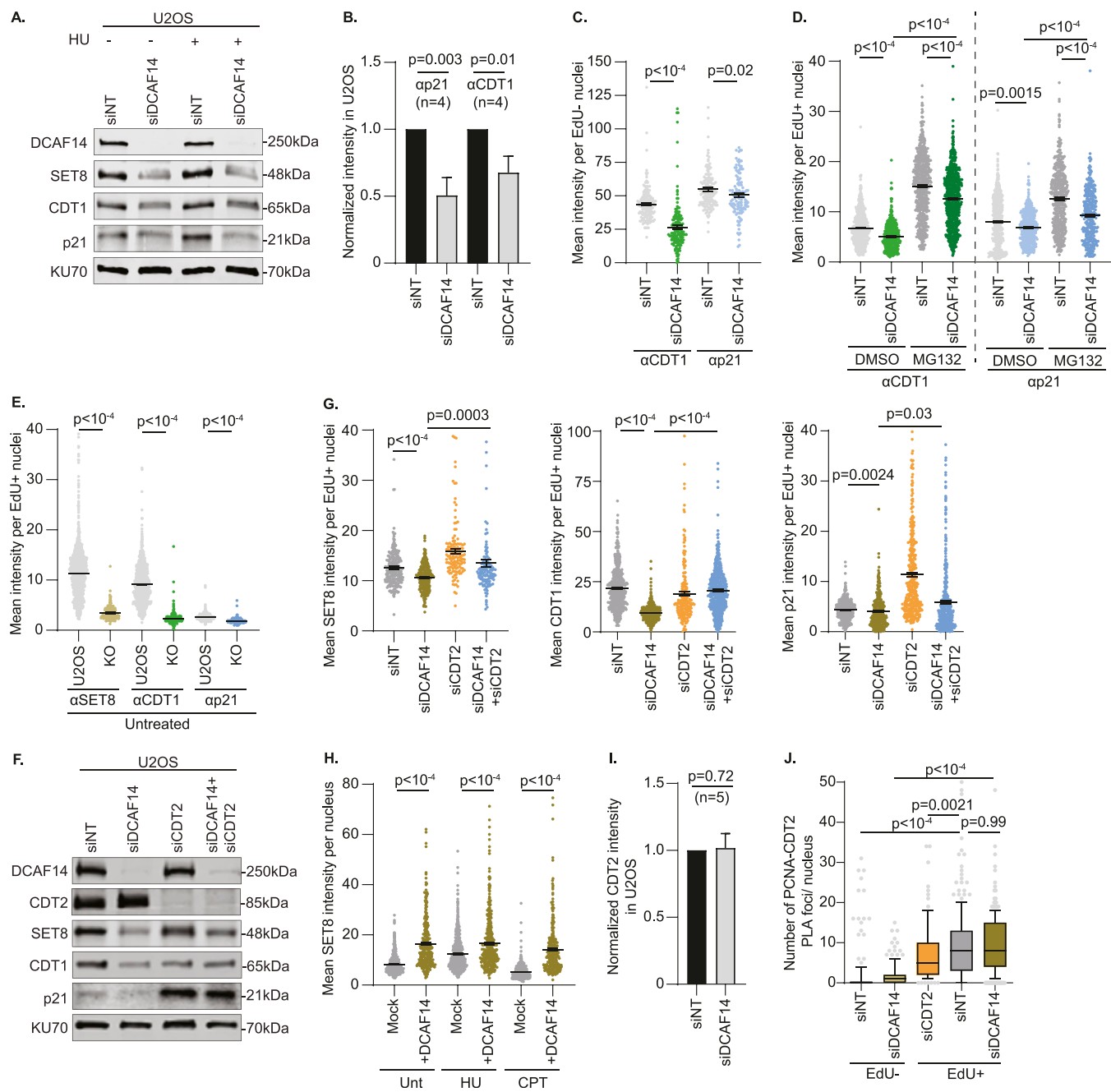

**Figure 3. CDT2 function is aberrantly elevated when DCAF14 is absent.**
**(A)** siNT- and siDCAF14-transfected U2OS cells were either left untreated or treated with HU for 4 h. Immunoblots were probed with the antibodies as shown. **(B)** Whole-cell lysates were extracted from siNT- and siDCAF14-transfected U2OS cells to analyze changes in p21 and CDT1. Graph represents normalized p21 and CDT1 intensities from four biological replicates. **(C)** siNT- and siDCAF14-transfected U2OS cells were pulsed with EdU for 30 min and immunostained for either CDT1 or p21 after preselecting EdU− nuclei. Graphs represent mean ± SEM using at least 200 nuclei. **(D)** siNT- and siDCAF14-transfected U2OS cells were pretreated with either DMSO or MG132 for 2 h and pulsed with EdU during the last 30 min of treatment. Cells were immunostained for either CDT1 or p21 and mean nuclei intensity was measured by quantitative imaging after preselecting EdU+ nuclei. Graphs represent mean ± SEM using at least 500 nuclei. **(E)** Parental and DCAF14 KO U2OS cells were pulsed with EdU and immunostained for SET8, CDT1 or p21. Mean nuclei intensity was measured by quantitative imaging after preselecting EdU+ nuclei. Graphs represent mean ± SEM using at least 250 nuclei. **(F)** Whole-cell lysates were extracted from U2OS cells transfected with the indicated siRNAs. Immunoblots were probed with the antibodies as shown. **(G)** U2OS cells transfected with the indicated siRNAs were pulsed with EdU and immunostained for SET8, CDT1 or p21. Mean nuclei intensity was measured by quantitative imaging after preselecting EdU+ nuclei. Graphs represent mean ± SEM using at least 125 nuclei. **(H)** Mock or DCAF14 cDNA-transfected U2OS cells were left either untreated or treated with 4 mM HU or 1 μM CPT for 4 h before immunostaining for SET8. Mean nuclei intensity was measured by quantitative imaging DAPI-stained nuclei. Graphs represent mean ± SEM using at least 200 nuclei. **(I)** Whole-cell lysates were extracted from siNT- and siDCAF14-transfected U2OS cells to analyze changes in CDT2. Graph represents normalized CDT2 intensities from five biological replicates. **(J)** U2OS cells transfected with the indicated siRNAs were pulsed with EdU for 30 min and subjected to PLA analyses using antibodies targeting CDT2 and PCNA. Number of PLA foci was measured after preselecting EdU+ and EdU− nuclei. Box and whisker plots represent 10–90 percentile using at least 100 nuclei.
Source data are available for this figure.

camptothecin (CPT) treatment (Figs 3H and S2D). We conclude that CDT2-mediated repression of PIP-degron substrates is aberrantly elevated when DCAF14 is absent.

To further support this conclusion, we sought to determine the mechanistic basis for altered CDT2 function in DCAF14-deficient conditions. Steady-state levels of CDT2 remain unaltered in DCAF14-depleted and DCAF14-deleted cells (Figs 3F and I and S3C) indicating that CDT2 itself is not proteolytically targeted by DCAF14. Replication-coupled destruction by CRL4$^{CDT2}$ occurs on PCNA occupying replication forks. Thus, we tested for differences in proximity of CDT2 to PCNA in replicating cells by proximity ligation assay (PLA) analysis and detect similar degree of PCNA-CDT2 PLA foci in DCAF14-proficient and DCAF14-deficient cells (Fig 3J). Thus, the relative cellular abundance and fork occupancy of CDT2 is largely unchanged when DCAF14 is silenced. Consistent with our results above (Fig S1D), acute exposure to neddylation inhibitor MLN4924 eliminates the slower-migrated species of neddylated CUL4A (Fig S2E) and stabilizes CDT1 and p21 in both unperturbed conditions and in presence of added exogenous stress (Fig S2F and G). We conclude that DCAF14 loss increases CRL4$^{CDT2}$ activity, reflecting enhanced down-regulation of CDT2 substrates.

## Elevated CDT2 function impairs stalled fork stability in DCAF14-deficient cells

DCAF14 absence renders stalled replication forks to undergo breakage thereby causing genome instability (Townsend et al, 2021). Excessive down-regulation of CDT2 substrates after DCAF14 inactivation suggested that stalled fork instability might be a direct consequence of aberrant CDT2 function. To test this hypothesis, we first asked whether replication elongation is modified in unchallenged, DCAF14-depleted cells. DNA-combing assays indicate that fork speeds remain unaltered in DCAF14-deficient cells, with no significant increase in replication fork asymmetry (Fig 4A). This observation is consistent with previous findings using DNA fiber spreading (Townsend et al, 2021) and is expected since CRL4$^{CDT2}$ triggers substrate proteolysis at active replication forks in an unperturbed S-phase. In contrast, we speculated that removing CDT2 should ameliorate replication fork instability in DCAF14-depleted cells that are challenged with replication stress. To test this idea, we first assessed whether CDT2 contributes to nascent DNA stability upon fork stalling. CDT2 depletion does not affect fork elongation rates in the presence of low-dose CPT across various cell types (Fig S3A and B). Furthermore, when CDT2-deficient cells are challenged with high-dose HU for 4 h, nascent DNA stability remains unaffected (Fig 4B). Next, we silenced CDT2 in both siDCAF14-transfected and DCAF14 KO cells. Strikingly, nascent strand degradation in DCAF14 absence is suppressed upon removal of CDT2 (Figs 4B and S3C and D). This result is not due to off-target effects of the siRNA targeting CDT2 because similar rescue is also observable using an independent siRNA (Fig S3E and F). Furthermore, cullin inhibitor MLN4924 also restores replication fork protection suggesting that CDT2 proteolytic activity triggers nascent DNA digestion in DCAF14-depleted cells (Fig 4C). These data demonstrate a function of DCAF14 in modulating CRL4$^{CDT2}$-mediated turnover to promote nascent DNA stability.

When DCAF14-deficient cells are exposed to replication stress, stalled forks fail to recover because of collapse into DSBs (Townsend et al, 2021). Because CDT2 depletion reverts nascent DNA instability in cells devoid of DCAF14, we asked whether replication stress-associated genome instability in DCAF14-silenced cells is a consequence of unregulated CDT2 function. Using neutral comet assay, we find that elevated DSB accumulation is reversed when CDT2 is inactivated in CPT-treated, DCAF14-deficient cells (Fig 4D). Moreover, fork recovery after prolonged stalling with HU is increased after CDT2 depletion in DCAF14-silenced cells (Fig 4E). We conclude that the aberrant increase in CDT2 activity leads to collapse of stalled replication forks in DCAF14-deficient cells.

## SET8 mediates replication fork protection

The data so far illustrate a model whereby the stalled fork instability events in DCAF14-deficient cells arise from excessive proteolytic activity of CDT2. Previous studies depict continued engagement of CDT2 in proteolyzing PIP-degron substrates when cells are challenged with genotoxic stress (Panagopoulos et al, 2020). However, before being subjected to proteasomal degradation, several CDT2 substrates transiently act at damage sites and stalled forks to regulate DNA repair (Oda et al, 2010; Bacquin et al, 2013; Tsanov et al, 2014). Thus, we investigated whether protection of nascent DNA is regulated by CDT2 substrates. To test this idea, we assessed the stability of newly synthesized DNA after prolonged exposure to HU in cells transiently depleted of SET8, p21, and CDT1. Fork protection remains intact in cells transiently depleted of CDT1 and p21 using two independent siRNAs (Figs 5A and S4A–C). In comparison, silencing SET8 using siRNAs that target either the open reading frame or 3′ UTR causes nascent strand degradation (Figs 5B and S4D). To rule out any possible off target effects because of siRNA-mediated depletion of SET8, we used the ability of SET8 inhibitor UNC0379 to acutely reduce steady-state levels of SET8 (Fig 1C). Remarkably, HU-stalled forks are also prone to nascent DNA digestion when exposed to UNC0379 for 4 h (Fig 5B). These data demonstrate a role of SET8 in protecting nascent DNA at stalled replication forks. To further support this conclusion, we monitored engagement of SET8 at replication forks using PLA-based SIRF assay (Roy et al, 2018) by quantifying proximity of SET8 to EdU-labeled DNA. In both unperturbed and HU-treated cells, we detect a significant presence of SET8-Biotin PLA foci which is diminished in SET8-deficient conditions (Fig 5C). Thus, SET8-mediated replication fork protection corresponds to the relative occupancy of SET8 at replication forks.

Replication fork remodeling of stalled forks into four-way intermediates is a common prerequisite for nuclease-mediated digestion of nascent DNA (Bhat & Cortez, 2018; Rickman & Smogorzewska, 2019). To determine whether nascent DNA digestion in SET8-depleted cells is dependent on fork reversal enzymes, we co-depleted either SMARCAL1 or ZRANB3 in SET8-deficient cells. Inactivating either SMARCAL1 or ZRANB3 is sufficient to reverse nascent DNA digestion in SET8-silenced cells (Figs 5D and S4E). Thus, fork reversal is a likely prerequisite for fork degradation when SET8 is absent. Next, we sought to determine the nuclease responsible for digesting nascent DNA in the absence of SET8. For this purpose, we used small-molecule inhibitors mirin and C5 to block nuclease activities of MRE11 and DNA2, respectively. Fork protection is restored in SET8-depleted cells upon MRE11 or DNA2 inhibition (Fig 5E) suggesting that both nucleases MRE11 and DNA2 contribute to

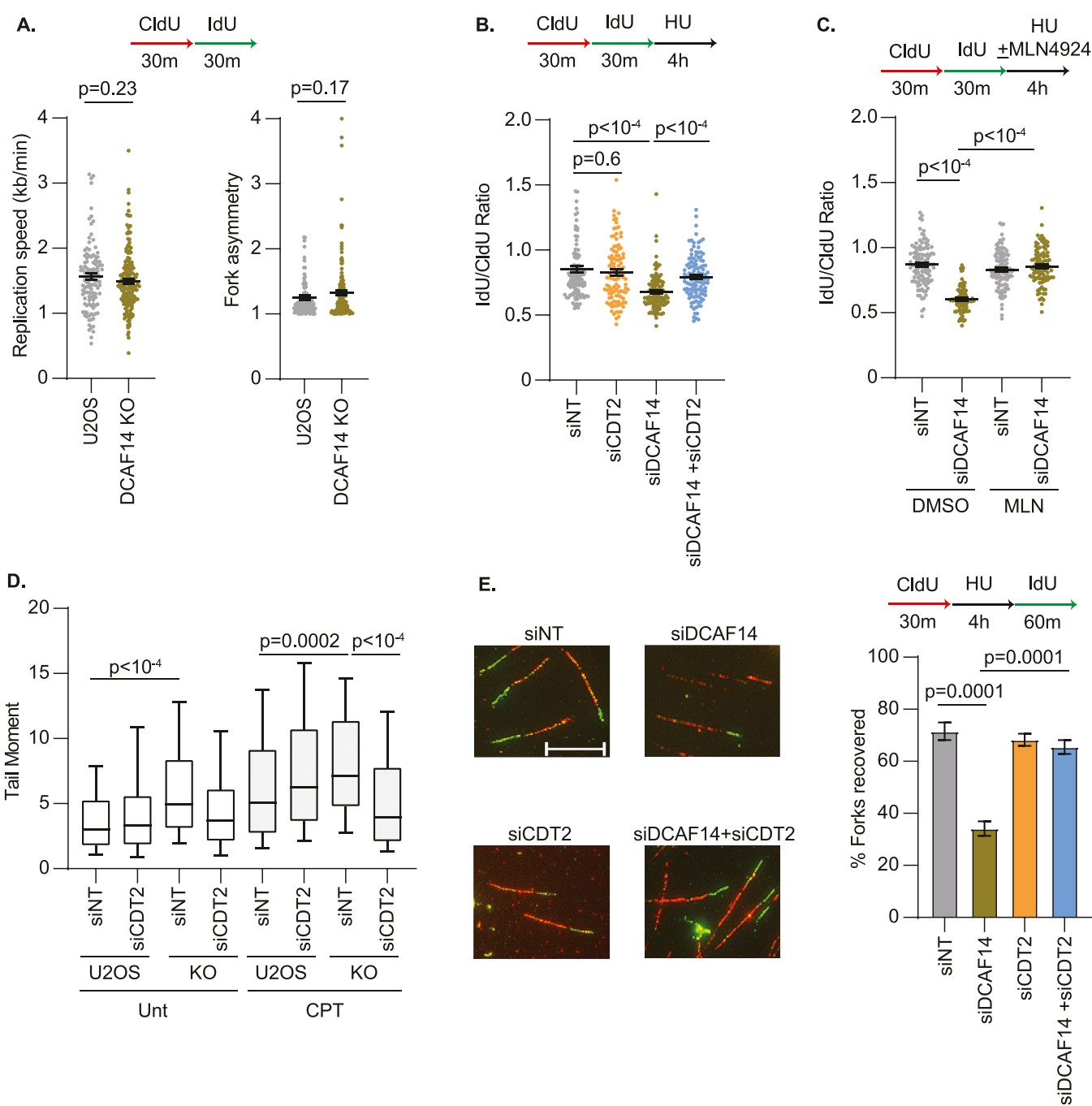

**Figure 4.  CDT2 impairs stalled fork stability in DCAF14-deficient cells.**
**(A)** U2OS or DCAF14 KO cells were pulse-labeled with CldU for 30 min followed by IdU for 30 min before DNA combing analysis to measure replication fork speed and replication fork asymmetry. **(B)** U2OS cells transfected with the indicated siRNAs were subjected to fork protection analysis. **(C)** siNT- and siDCAF14-transfected U2OS cells were subjected to fork protection analysis. DMSO or MLN4924 were added concomitantly with HU for 4 h. **(D)** Tail moments were measured for parental and DCAF14 KO U2OS cells transfected with the indicated siRNAs using neutral comet assay. Cells were either untreated or treated with CPT for 1 h. Box and whisker plots represent 10–90 percentile using at least 100 nuclei. **(E)** Stalled fork recovery post 4 h of HU treatment was measured for the U2OS cells transfected with the indicated siRNAs using DNA fiber analysis. Scale bar = 10 $\mu$m. Graphs represent mean ± SD from three biological replicates. Fiber labeling experiments depict mean ± SEM and at least 100 fibers were quantified.
Source data are available for this figure.

the digestion of nascent DNA in SET8-deficient cells. To further corroborate this result, we performed siRNA-mediated depletion of DNA2 and MRE11 in SET8-deficient cells and observe restoration of fork protection by depleting either MRE11 or DNA2 (Fig S4F and G). These results indicate that loss of SET8 triggers MRE11- and DNA2-dependent digestion of nascent DNA at remodeled replication forks.

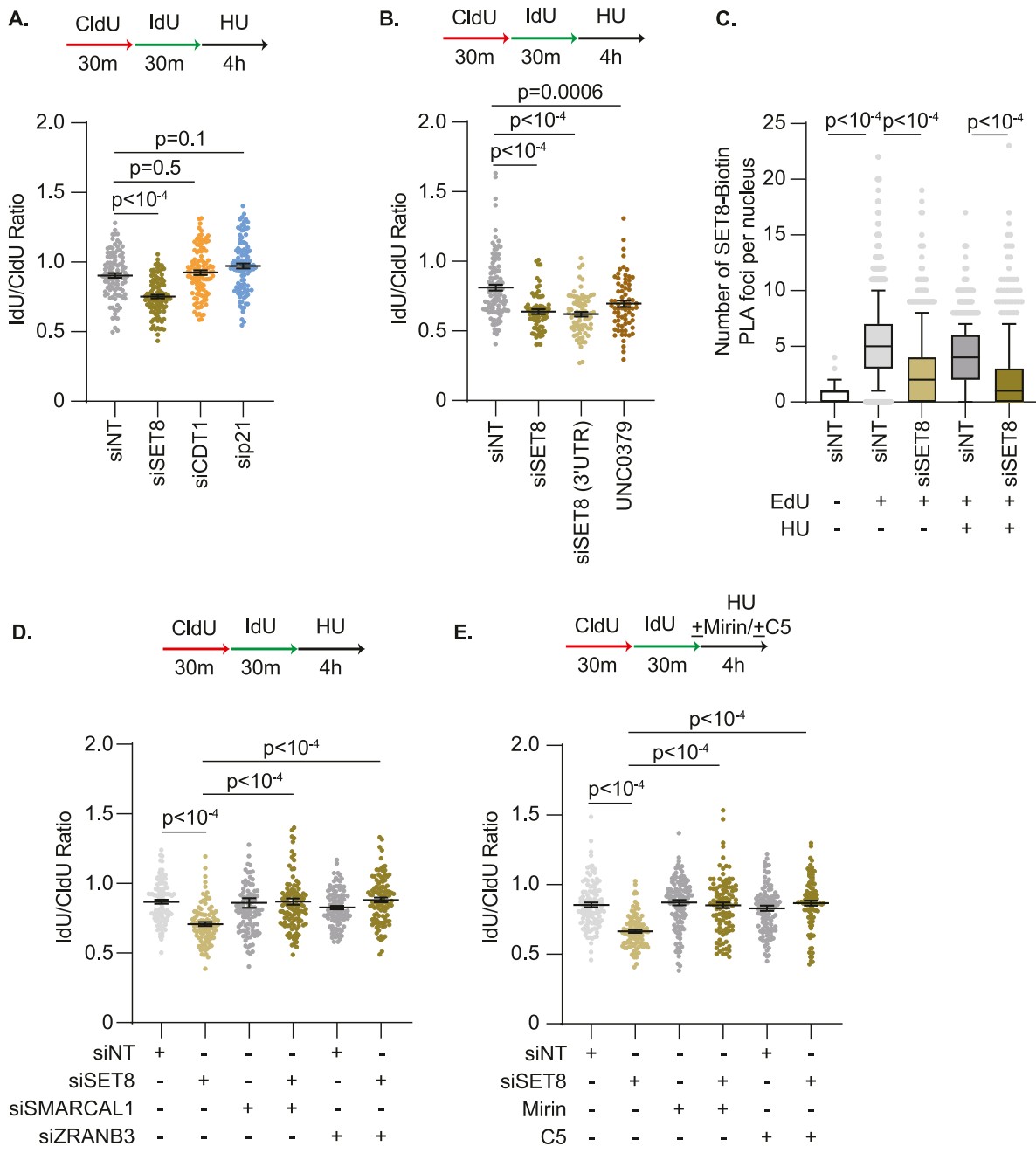

**Figure 5. SET8 mediates replication fork protection.**
**(A)** U2OS cells transfected with the indicated siRNAs were subjected to fork protection analysis. **(B)** U2OS cells either transfected with the indicated siRNAs or concomitantly treated with HU and UNC0379 for 4 h were subjected to fork protection analysis. **(C)** Untreated or HU-treated siNT- and siSET8-transfected U2OS cells were subjected to SIRF analyses using antibodies targeting SET8 and biotin. Box and whisker plots represent 10–90 percentile using at least 450 nuclei. **(D)** U2OS cells transfected with the indicated siRNAs were subjected to fork protection analysis. **(E)** U2OS cells transfected with the indicated siRNAs were subjected to fork protection analysis. MRE11 inhibitor mirin or DNA2 inhibitor C5 were added concomitantly with HU for 4 h where indicated. Fiber-labeling experiments depict mean ± SEM and at least 100 fibers were quantified.
Source data are available for this figure.

## DCAF14 promotes SET8-dependent replication fork protection

The dependency on fork reversal enzymes SMARCAL1 and ZRANB3, and nucleases MRE11 and DNA2, for nascent strand degradation in SET8-deficient cells is identical to DCAF14-deficient cells (Townsend et al, 2021) suggesting that DCAF14 and SET8 function in the same pathway to mediate replication fork protection. Thus, we hypothesized that unregulated degradation of SET8 by CRL4$^{CDT2}$ triggers nascent DNA digestion in the absence of DCAF14. To test this prediction, we first asked whether proteasomal inhibition alleviates

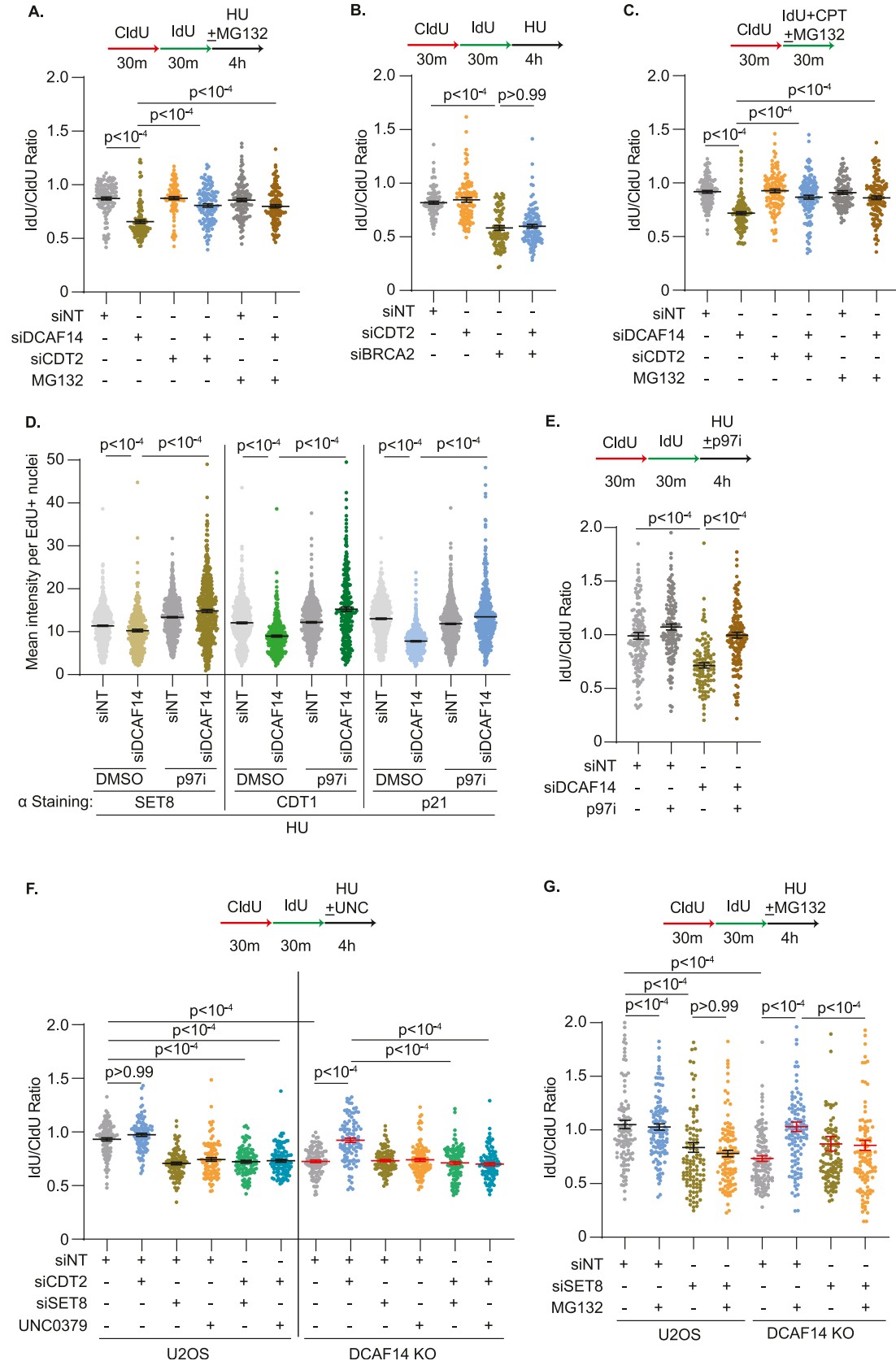

nascent strand degradation in DCAF14-deficient cells. HU-induced fork degradation in DCAF14-depleted cells is suppressed with concomitant exposure to proteasomal inhibitor MG132, similar to the rescue observed in CDT2 co-depleted cells (Figs 6A and S5A). Unlike DCAF14-depleted cells, CDT2 depletion is unable to restore fork protection in BRCA2-deficient cells (Figs 6B and S5B) indicating that CDT2-dependent nascent strand degradation is generated in conditions of DCAF14 deficiency. In addition, when DCAF14-deficient cells are challenged with low dose CPT, the curtailed fork speeds are rescued by either co-depleting CDT2 or MG132 treatment (Fig 6C). This is not due to alterations in fork speeds in unperturbed conditions (Fig S5C). Thus, in the absence of DCAF14, excessive CDT2-mediated proteolysis causes degradation of nascent DNA.

To further corroborate these observations, we investigated the contribution of p97 in CDT2-dependent nascent DNA digestion. p97 ATPase, a central component of the ubiquitin–proteasome system, extracts CDT2-induced polyubiquitinated CDT1 and SET8 from chromatin in response to UV damage (Raman et al, 2011). We speculated that nascent DNA digestion in DCAF14-deficient cells consequentially arises from the coordinated actions of CDT2 and p97. To test this idea, we first isolated chromatin fractions from p97i-treated DCAF14-proficient and DCAF14-deficient cells and observe increased accumulation of ubiquitinated substrates compared with control cells (Fig S5D). Next, we determined whether p97 inhibition rescues CDT2 substrate levels in HU-treated, DCAF14-deficient cells by immunofluorescence analyses. Insoluble levels of SET8, CDT1, and p21 are increased in EdU-positive nuclei of HU-treated, DCAF14-deficient cells after p97 inhibition for 4 h (Fig 6D) demonstrating a role of p97 in mediating turnover of CDT2 substrates in DCAF14-inactivated cells. Because inactivating CDT2 or p97 restores levels of CDT2 substrates, we predicted that acute inhibition of p97 should restore fork protection in DCAF14-deficient cells. Indeed, nascent DNA digestion is abolished when DCAF14-deficient cells are concomitantly treated with HU and p97i (Fig 6E). These results indicate that reversing turnover of CDT2 substrates from chromatin can restore replication fork protection in DCAF14-silenced cells.

Because SET8 promotes replication fork protection, we measured SET8 association with EdU-labelled DNA using SIRF analyses to determine whether DCAF14 absence lowers SET8 presence at replication forks. In both unperturbed and HU-treatment conditions, SET8-Biotin PLA foci are reduced in DCAF14-deficient cells (Fig S5E). Importantly, SET8 abundance at forks is rescued when CDT2 is co-depleted. These results suggest that restoring SET8 levels at forks abrogates nascent strand degradation in DCAF14-depleted cells. Accordingly, removing SET8 should negate restoration of fork protection in DCAF14-deficient cells that lack functional CDT2. To

test this prediction, we transiently suppressed SET8 in CDT2-depleted parental U2OS and DCAF14 KO cells. CDT2 depletion does not trigger nascent strand degradation in parental U2OS cells as expected. However, concomitant loss of SET8 either by siRNA transfection or UNC0379 treatment triggers digestion of nascent DNA in CDT2-depleted cells suggesting that SET8 mediates replication fork protection when CDT2 is absent (Figs 6F and S5F). Strikingly, whereas CDT2 depletion restores fork protection in DCAF14 KO cells, simultaneous loss of SET8 reverts this rescue further indicating that restoring SET8 levels alleviates nascent DNA digestion in DCAF14-deficient cells. Consistent with this interpretation, MG132-dependent fork protection in DCAF14 KO cells is abolished upon SET8 co-depletion (Figs 6G and S5G). Overall, these data indicate that DCAF14 prevents uncontrolled CDT2-dependent proteolysis of SET8 to mediate replication fork protection.

## Discussion

In this study, we find that DCAF14 promotes stalled fork stability by regulating CRL4$^{CDT2}$ activity. When DCAF14 is absent, CDT2 function is aberrantly elevated causing nascent strand degradation and collapse of stalled replication forks into DSBs. Specifically, CRL4$^{CDT2}$ proteolytic activity is up-regulated because CDT2 substrates undergo increased turnover in DCAF14-deficient cells. Strikingly, excessive loss of SET8 triggers nascent strand degradation. Moreover, restoration of fork protection by CDT2 removal in DCAF14-deficient conditions is suppressed when SET8 is depleted. We postulate that DCAF14 antagonizes CDT2 function transiently at stalled replication forks. By this mode of action, PCNA-associated SET8 mediates replication fork protection before degradation by CRL4$^{CDT2}$. Although CRL4$^{CDT2}$-dependent proteolysis is essential during S-phase in genome maintenance, restraining this function is necessary for genome integrity in response to fork stalling events.

We propose the following model for DCAF14 actions as a stress-response protein (Fig S5H). DCAF14 is recruited to stalled replication forks via hitherto unknown mechanism. Upon enrichment, DCAF14 limits the activity of CRL4$^{CDT2}$ to allow transient stabilization or local accumulation of CDT2 substrate SET8 at stalled sites of DNA synthesis. CRL4$^{CDT2}$-mediated proteolysis is essential during unperturbed S-phase and in conditions of genotoxic stress to prevent re-replication. Thus, although CDT2 substrates continue to be degraded in a PCNA-dependent manner, we speculate that the association of SET8 at replication forks is modulated by the antagonistic actions of DCAF14 thereby enabling SET8 to participate in replication

**Figure 6. CDT2-dependent down-regulation of SET8 causes nascent strand degradation in DCAF14-deficient cells.**
**(A)** U2OS cells transfected with the indicated siRNAs were subjected to fork protection analysis. MG132 was added concomitantly with HU for 4 h where indicated. **(B)** U2OS cells transfected with the indicated siRNAs were subjected to fork protection analysis. **(C)** U2OS cells transfected with the indicated siRNAs were subjected to fork elongation assays in the presence of 100 nM CPT as shown. MG132 was added concomitantly with CPT for 30 min where indicated. **(D)** siNT- and siDCAF14-transfected U2OS cells were pulsed with EdU followed by treatment with HU for 4 h, concomitant with DMSO or p97i as indicated, and subjected to immunofluorescence analyses. Mean nuclei intensity of SET8, CDT1 or p21 was measured by quantitative imaging after preselecting EdU+ nuclei. Graph depicts mean ± SEM using at least 300 nuclei. **(E)** U2OS cells transfected with the indicated siRNAs were subjected to fork protection analysis. p97i was added concomitantly with HU for 4 h where indicated. **(F)** Parental and DCAF14 KO U2OS cells transfected with the indicated siRNAs were subjected to fork protection analysis. UNC0379 was added concomitantly with HU for 4 h where indicated. **(G)** Parental and DCAF14 KO U2OS cells transfected with the indicated siRNAs were subjected to fork protection analysis. MG132 was added concomitantly with HU for 4 h where indicated. Fiber-labeling experiments depict mean ± SEM and at least 100 fibers were quantified.
Source data are available for this figure.

fork protection. Indeed, after recruitment by PCNA, SET8 mediates 53BP1 focal accumulation at DSB sites before being degraded by CRL4[CDT2] (Oda et al, 2010). How does SET8 prevent nascent strand degradation? SET8 is the sole monomethyltransferase for H4K20. Studies demonstrate that H4K20me1 is the primary determinant for 53BP1 recruitment in NHEJ-dependent DSB repair (Dulev et al, 2014) and this function may facilitate 53BP1-mediated replication fork protection as previously described (Liu et al, 2020). H4K20me1 also promotes chromatin compaction in mitosis (Centore et al, 2010) and this function may permit localized chromatin compaction on regressed arms to prevent nuclease-dependent digestion of nascent DNA. In this respect, alterations in nucleosome remodeling and chromatin accessibility can directly impact stalled fork stability (Thakar & Moldovan, 2021). Although H4K20me1 levels are suppressed in SET8-inactivated cells, a direct causative role for this methylation event in stalled fork stability needs further investigation.

Our results demonstrate that blocking proteasome by MG132 or inhibiting cullins by MLN4924 transiently up-regulates SET8 in both replicating and nonreplicating cells in DCAF14-deficient conditions. In addition to CRL4[CDT2], SET8 is also a proteolytic target of APC[CDH1] during mitosis (Wu et al, 2010) and SCF[β−TRCP] in G1-phase (Wang et al, 2015). Heightened substrate down-regulation observed in DCAF14-deficient cells outside of S-phase could be due to the cumulative effects of excessive CDT2 proteolytic activity during S-phase or regulation by DCAF14 that is replication-independent. In comparison, several observations support our conclusion that DCAF14 negatively regulates CDT2 function during DNA replication. First, acute treatment of DCAF14-deficient cells with MG132, p97i, and MLN4924 rescues CDT2 substrate levels in EdU+ nuclei. Second, CDT2 depletion increases CDT2 substrates in DCAF14-depleted cells. Third, silencing CDT2 or acute exposure to MG132, p97i or MLN4924 restores fork protection at HU-stalled forks in DCAF14-deficient cells. Fourth, CDT2 depletion enables stalled fork restart when DCAF14 is absent. Fifth, restoration of fork protection in cells co-depleted of DCAF14 and CDT2 is abrogated when SET8 is removed.

CRL4[CDT2] proteolytic degradation during DNA replication is essential to promote cell cycle progression and genome stability. Absence of CDT2 results in re-replication because of excessive stabilization of proteolytic substrates CDT1, SET8, and p21 highlighting the need for uninterrupted CDT2 activity during S-phase. In contrast, CDT2 activity is also suppressed to regulate key cellular transition events. CDT2 phosphorylation by CDK1 suppresses CDT2 recruitment to chromatin during G2/M transition to allow SET8 reaccumulation and chromatin compaction for mitosis (Rizzardi et al, 2015). CDT2 is also targeted for proteasomal degradation by ubiquitin ligase complex SCF[FBXO11] to regulate cell cycle exit and differentiation (Rossi et al, 2013). Our studies reveal that CRL4[CDT2] activity is negatively regulated at stalled replication forks to promote replication fork stability. Steady-state levels of CDT2 are minimally altered when DCAF14 is depleted or overexpressed indicating that DCAF14 does not target CDT2 for proteolytic degradation. This prediction is consistent with the crucial role of CDT2 in S-phase because eliminating CDT2 can be detrimental for genome stability while cells are still in the process of completing genome duplication. We predict that DCAF14 fine-tunes CDT2 activity at stalled replication forks to allow controlled turnover of SET8 and thereby mediate fork protection before its degradation by CRL4[CDT2].

PCNA molecules on replication forks serve as a platform for assembly of CRL4[CDT2] that targets PIP-degron–containing substrates for polyubiquitination and subsequent degradation. Because loss of CRL4 components DDB1 and CUL4B also elicit nascent strand degradation (Townsend et al, 2021), DCAF14 likely subverts CDT2 activity as a component of CRL4 complex. Recent studies demonstrate that cellular DCAFs exist as DDB1 bound pools that are constantly sampled by CRL complexes (Reichermeier et al, 2020). The cellular abundance of cullin scaffold is lower compared to the DDB1-DCAF pool illustrating the limited availability of cullins to be simultaneously engaged in substrate ubiquitination. In support of these observations, we find that DCAF14 overexpression increases SET8 levels suggesting CRL4 components are perhaps sequestered by DDB1–DCAF14 complexes to block CDT2 activity. In contrast, DCAF14 absence allows CDT2 actions to be aberrantly elevated. In support of this model, inhibiting CRL4[CDT2] activity using MLN4924 prevents proteolysis of CDT2 substrates and alleviates fork problems in DCAF14-deficient cells. Thus, increased CDT2 activity in DCAF14-deficient conditions is favorable in unperturbed conditions but detrimental in conditions of replication stress. Whether CRL4[CDT2] and CRL4[DCAF14] complexes coexist at stalled forks or function as codependent exchangeable entities remain to be investigated. An alternate possibility is that SET8 could directly be modulated by CRL4[DCAF14] in a proteolytic-independent mechanism. Although this model cannot be excluded, we find that multiple substrates are subjected to increased CDT2-driven turnover in DCAF14 deficiency indicating a role of DCAF14 in modulating CDT2 activity. In summary, we find a negative regulatory function of DCAF14 in controlling CDT2 activity to mediate replication fork protection. This restraint action permits SET8 to act in a transient manner to suppress nascent strand degradation and promote replication fork stability.

# Materials and Methods

### Cell culture

U2OS and HeLa cell lines were cultured in DMEM media with 7.5% FBS. hTERT-RPE1 cell lines were cultured in DMEMF12 with 7.5% FBS. All cell lines were incubated at 37°C with 5% CO2. DCAF14 KO U2OS cell lines used in this study were generated as described previously (Townsend et al, 2021).

### Transfections

Plasmid DNA transfections were achieved using FUGENE HD (Promega) in U2OS cells. siRNA transfections were performed using the following reagents: Dharmafect1 (Dharmacon) was used for U2OS cells, RNAimax (Thermo Fisher Scientific) was used for hTERT-RPE1 and HeLa cells. All assays were performed 72 h post transfection.

### Immunofluorescence

For immunofluorescence (IF) analyses, detergent extraction was achieved using 0.5% Triton X-100 before fixation with either 3% paraformaldehyde/2% sucrose or 4% paraformaldehyde. After

blocking with 5% BSA/PBS, cells were incubated with primary and secondary antibodies. Coverslips were mounted using Prolong Gold with DAPI (Invitrogen). To detect replicating cells in conditions without added HU, cells were pulsed with 10 $\mu$M EdU for 30 min. For MLN4924 experiments, cells were pretreated with MLN4924 for 4 h and EdU was added during the last 60 min of treatment. For MG132 experiments, cells were pretreated with 2 h of MG132 and EdU was added during the last 30 min of treatment. For all HU treatments, cells were prelabeled for 10 min with EdU followed by 4 h exposure to HU with or without MLN4924, MG132 and p97i. Genome-incorporated EdU was detected by click chemistry using Alexa Fluor 488 or 594 azide. For PLAs, Duolink In Situ Mouse/Rabbit Kit (Sigma-Aldrich) was used. For in situ Protein Interaction with Nascent DNA Replication Forks (SiRF) assay, cells were labeled with 125 $\mu$M EdU for 10 min before fixation. Image acquisition was performed on a Keyence BZ-X810 fluorescence microscope using a 20X objective (0.75NA). Cell profiler or BZ-X800 analyzer was used to measure mean nuclei intensity. Macro cell count on Keyence analysis software was used to quantify PLA foci. All measurements of mean nuclei intensities are depicted in arbitrary units.

### Fiber analysis by DNA spreading

DNA fiber labeling analysis was performed as described previously (Jackson & Pombo, 1998). To measure fork speeds in unchallenged conditions, cells were pulsed sequentially with 20 $\mu$M CldU and 100 $\mu$M IdU for 30 min each. To analyze fork elongation in the presence of CPT, cells were pulsed with CldU for 30 min, followed by IdU for 30 min in the presence of 100 nM CPT. MG132 was added concomitantly with CPT for 30 min where indicated. For nascent strand degradation assays, cells were sequentially pulsed with CldU and IdU for 30 min each followed by release into 4 mM HU for 4 h. MG132, p97i, MLN4924, UNC0379, MRE11i (Mirin), and DNA2i (C5) treatments in NSD analyses were performed in the presence of HU for 4 h. To assess fork recovery after release from replication stress, cells were pulsed with CldU for 30 min, released into 4 mM HU for 4 h, and pulsed with IdU for 60 min. After treatments and labeling schemes, cells were harvested, lysed on slides and DNA was stretched by tilting the slides. DNA was then fixed using 3:1 solution of methanol:acetic acid and stored at −20°C overnight. Next day, DNA was denatured using 2.5 N HCl for 45 min, blocked in PBS containing 0.1% Triton X-100 and 10% goat serum for 1 h, stained with primary antibodies for 2 h to recognize IdU and CldU followed by staining with secondary antibodies for 1 h. Images were acquired using a 60X oil objective (1.4 NA) on Keyence BZ-X710 and fiber lengths were analyzed using ImageJ.

### DNA combing assay

Molecular combing was performed following Genomic Vision manufacturer's instructions. Briefly, cells were sequentially labeled with CldU and IdU for 30 min, harvested, and embedded in low-melt agarose plugs. Plugs were treated overnight with Proteinase K, washed thoroughly, and treated overnight with beta-agarase. After agarase treatment, free DNA was transferred to reservoir containing MES buffer at pH 5.5 and allowed to acclimate to room temperature before combing with GenomicVision coverslips and FiberComb

apparatus. Combed coverslips were held at 60°C for 2 h before proceeding with immunostaining. Slides were scanned using FiberVision S automated scanner (Genomic Vision). 40X objective magnification was used with 8-bit depth for each channel. Data were extracted and analysis of replication dynamics was performed using FiberStudio 2.0 software (Genomic Vision).

### Neutral comet assay

DNA DSBs were detected using Trevigen comet assay kit. Tail moments were scored using Comet Score software (Tritek) and data presented as box and whisker plots.

### Flow cytometry

For cell cycle analyses, U2OS cells were labeled with 10 $\mu$M EdU for 30 min. After trypsinization, cells were fixed in 70% ethanol overnight. Click reaction was then performed to conjugate EdU to AlexaFluor azide 488 for 30 min. Cells were washed and resuspended in 3 $\mu$M DAPI (1 $\mu$g/ml). Cells were harvested and filtered into a flow cytometry tube for flow cytometry analyses. Analyses were carried out on BD FACSCanto II.

### Whole-cell lysate and chromatin extraction

To obtain whole-cell lysates, cell pellets were incubated on ice for 30 min in RIPA lysis buffer (50 mM Tris-Cl pH = 7.4, 1% NP-40, 150 mM NaCl, 0.1% SDS, 0.5% sodium deoxycholate, 1 mM sodium ortho-vanadate, 1 mM aprotinin, 1 mM leupeptin, 1 mM DTT, 1 mM PMSF) supplemented with Pierce universal nuclease and 1 mM $MgCl_2$. After centrifugation for 30 min, supernatant was isolated and quantified before immunoblotting analyses. Chromatin extractions were performed as previously described (Mendez & Stillman, 2000) with an overnight incubation step in Pierce universal nuclease (Thermo Fisher Scientific).

### Protein turnover assays

For assessment of protein turnover, U2OS cells were treated with either 10 $\mu$M MG132 for the indicated times. Cell lysates were harvested using RIPA lysis buffer and immunoblots were probed with the indicated antibodies.

### Quantification and statistical analysis

All statistical analyses were completed using Graphpad Prism 9 (Graphpad). For pairwise comparisons within the same experiment, Kruskal–Wallis one-way ANOVA test was used followed by Dunn's multiple comparisons test to calculate $P$-value. A two-tailed Mann–Whitney test was used for all PLA analyses. Unpaired $t$ test was used in immunoblotting experiments. Significance values were derived using $P$-value = 0.05 as cutoff. Statistical details can be found in the figure panels and legends including sample size, definition of center and dispersion measures, and $P$-values. All experiments were performed at least twice and representative experiments are shown, unless otherwise indicated.

## Antibodies

Antibodies used for Western blotting (WB) and immunofluorescence (IF) were used as described: DCAF14 (1:500, WB, NBP2-33883; Novus), CDT2 (1:500, WB, ABS500; Millipore), CDT1 (1:500, WB, 8064S; CST), SET8 (1:500, WB, 06-1304; Millipore and 1:200, IF, 2996T; CST), P21 (1:500, WB and 1:200, IF, sc-6246; SantaCruz), PHF8 (1:2,000, WB, A301-772A; Bethyl Laboratories), PCNA Rb (1:2,000, WB and 1:200, IF, ab18197; Abcam), PCNA Ms (1:200, IF, sc-56; SantaCruz), CUL4A (1:250, WB, ab72548; Abcam), MRE11 (1:500, WB, 4895S; CST), DNA2 (1:500, WB, PA5-77943; Invitrogen), SMARCAL1 (1:500, WB, sc-376377; SantaCruz), ZRANB3 (1:500, WB, A303-033A; Bethyl Laboratories), BrdU-Rat (1:10, IF, ab6326; Abcam), BrdU-Ms (1:100, IF, 347580; BD Biosciences), Biotin Rb (1:200, IF, A150-109A; Bethyl Laboratories), Biotin Ms (1:200, IF, 200-002-211; Jackson ImmunoResearch), KU70 (1:2,000, WB, ab92450; Abcam), H3 (1:6,000, WB, ab6326; Abcam), H4 (1:4,000, WB, ab177840; Abcam), H4K20me1 (1:1 k, WB and 1:200, IF, 39729; Active Motif), H4K20me3 (1:1,000, WB, sc-134216; SantaCruz), H3K4me3 (1:1,000, WB, ab8580; Abcam), H3K9me1 (1:1,000, WB, ab176880; Abcam), H3K9me3 (1:1,000, WB, ab8898; Abcam), H3K27me3 (1:1,000, WB, ab6002; Abcam), H3K36me3 (1:1,000, WB, ab9050; Abcam), WDR5 (1:500, WB, sc-393080; SantaCruz), RBBP5 (1:500, WB, sc-271072; SantaCruz), BRCA2 (1:500, WB, OP95; Millipore Sigma), HA (1:1,000, WB, H3663; Sigma-Aldrich), donkey anti-rabbit 800CW (1:20,000, WB, 926-32213; LI-COR), donkey anti-mouse 800CW (1:20,000, WB, 926-32212; LI-COR), Alexa Fluor 594 goat anti-rabbit IgG (1:2,000, IF, A11037; Invitrogen), Alexa Fluor 594 goat anti-mouse IgG (1:2,000, IF, A11005; Invitrogen), Alexa Fluor 594 goat anti-rat IgG (1:2,000, IF, A11007; Invitrogen), Alexa Fluor 488 goat anti-mouse IgG (1:2,000, IF, A11029; Invitrogen). For DNA combing assay, the following antibodies were used: BrdU-Rat (1:305, IF, ab6326; Abcam), BrdU-Ms (1:6.25, IF, 347580; BD Biosciences), Ms anti-ssDNA (1:12.5, IF, AutoAnti-S; DSHB), goat anti-rat Cy5 (1:12.5, IF, ab6565; Abcam), goat anti-Ms Cy3 (1:12.5, IF, ab97035; Abcam), goat anti-Ms BV490 (1:12.5, IF, 115-685-166; Jackson).

## siRNAs and plasmids

ON-TARGETplus or custom siRNAs were obtained from Dharmacon except where indicated: DCAF14 (J-019291-06), CDT2 (L-020543-00-0005), CDT2#2 (GCUGAGCUUUGGUCCACUA), CDT1 (L-003248-00-0005), CDT1#2 (AACGUGGAUGAAGUACCCGACUU), SET8 (13067S, CST), SET8 (3′UTR) (Custom siRNA, Sense sequence: GAACA-GAUGGCCUUAUAUU, [Tanaka et al, 2017]), p21 (L-003471-00-0005), p21#2 (AACAUACUGGCCUGGACUG), DCAF14 (5′UTR) (pool of J-019291-07 and J-019291-08), MRE11 (J-009271-08), DNA2 (pool of D-026431-03 and D-026431-04, siGENOME Dharmacon), SMARCAL1 (J-013058-06), ZRANB3 (D-010025-03, siGENOME Dharmacon), WDR5 (L-013383-00-0005), RBBP5 (L-012008-00-0005), All-stars negative control siRNA (1027280; QIAGEN). Plasmids were obtained from Origene: DCAF14 cDNA clone (RC217114) and entry vector (PS100001). HA-tagged WT ubiquitin plasmid was obtained from Addgene (#17608; Plasmid).

## Compounds

The following drugs and inhibitors were used at the indicated concentrations: MG132 (10 $\mu$M), p97i (5 $\mu$M), HU (4 mM), CPT (100 nM for DNA fiber analyses and 1 $\mu$M for neutral comet assay), MLN4924 (1 $\mu$M), CSN5i-3 (1 $\mu$M), UNC0379 (10 $\mu$M), MRE11i (Mirin) (10 $\mu$M), and DNA2i (C5) (20 $\mu$M).

## Supplementary material

Fig S1 (related to Fig 2) demonstrates that DCAF14 regulates SET8 levels. Fig S2 (related to Fig 3) shows that CDT2-mediated proteolysis during S-phase is elevated in the absence of DCAF14. Fig S3 (related to Fig 4) demonstrates that CDT2 activity causes nascent strand degradation in DCAF14-deficient cells. Fig S4 (related to Fig 5) reveals a function of SET8 in replication fork protection. Fig S5 (related to Fig 6) demonstrates that regulation of CDT2 activity by DCAF14 enables SET8 to mediate stalled fork protection.

# Data Availability

Datasets were not generated in this study.

# Supplementary Information

# Acknowledgements

This work was supported by NIH R35GM137800 grant to H Dungrawala. We thank Charles Szekeres at the USF COM Fred Wright Jr Flow Cytometry Core facility for flow cytometry analyses. We are grateful to Ted Dawson for the pRK5-HA-Ubiquitin-WT vector.

## Author Contributions

N Tirado-Class: conceptualization, data curation, formal analysis, investigation, and writing—original draft, review, and editing.
C Hathaway: formal analysis and investigation.
A Nelligan: formal analysis and investigation.
T Nguyen: investigation.
H Dungrawala: conceptualization, formal analysis, supervision, funding acquisition, project administration, and writing—original draft, review, and editing.

## Conflict of Interest Statement

The authors declare that they have no conflict of interest.

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
