## [Reviewer comments · Life Science Alliance]

Life Science Alliance

DCAF14 regulates CDT2 to promote SET8-dependent replication fork protection

Neysha Tirado-Class, Caitlin Hathaway, Anthony Nelligan, Thuan Nguyen, and Huzefa Dungrawala

DOI: <https://doi.org/10.26508/lsa.202302230>

Corresponding author(s): Huzefa Dungrawala, University of South Florida

Review Timeline:

Submission Date:	2023-06-30
Editorial Decision:	2023-07-06
Revision Received:	2023-10-13
Editorial Decision:	2023-10-17
Revision Received:	2023-10-24
Accepted:	2023-10-26

Transaction Report:

Please note that the manuscript was previously reviewed at another journal and the reports were taken into account in the decision-making process at *Life Science Alliance*.

Reviews

Review #1

Comments to the Authors (Required):

This paper looks at the interplay of two Cul4 E3 ubiquitin ligases, CRL4DCAF14 and CRL4Cdt2, in regulating fork stability. The authors show that fork degradation in HU caused by DCAF14 knock-down is suppressed by also knocking down Cdt2 (or adding a CRL4 inhibitor), implying that CRL4Cdt2 is de-regulated in the absence of CRL4DCAF14. They then also present various lines of evidence that de-regulation of CRL4Cdt2 leads to loss of Set8, a known CRL4Cdt2 substrate in S phase. This leads to a model in which Set8 levels, which are normally low in S phase due to PCNA-coupled degradation by CRL4Cdt2, are lowered even further in the absence of CRL4DCAF14. In other words, even the low levels of Set8 that are normally present are doing something to preserve stalled forks because further reduction of Set8 is not tolerated. They also show evidence that in the absence of CRL4DCAF14, fork degradation involves for reversal and the nucleases DNA2 and MRE11.

This is a potentially interesting manuscript. However, much of the data, especially in the first half of the paper purporting to show Set8 destabilization in the absence of DCAF14, is not convincing, as outlined below. Also, the logic of the final experiments presented in Figure 6 is very convoluted. It would be simpler and more compelling to overexpress Set8 (WT, or a degron mutant), to show that this is sufficient to suppress the fork stability defects seen after DCAF14 loss. Finally, major aspects remain unexplained: how does Set8 protect fork stability? How does DCAF14 suppress CRL4Cdt2 activity?

Specific Points

1. Figure 1E is not convincing because there was clearly a Western transfer problem in the key, middle lane in the H4K20me1 blot. Also, H4 looks overloaded in lanes 3 and 4. Please redo and show lighter exposure of loading controls next time.
2. The lowered expression of SET8 in Figure 2A is not convincing because Ku70, which is presumably a loading control, is also lower in the siDCAF14 lanes. Hard to judge how much because again, blot is overexposed. SET8 levels need to be normalized to the loading control, but not when it's overexposed as in this figure. In Figure S1B, where Ku loading is equal, there is barely any reduction in SET8.
3. There is a dramatic drop in G2 population in U2OS cells in Figure S1A. Why is this?

4. To make the claim that the rescue of SET8 levels by MG132 is "coupled to DNA synthesis" they would need to show that MG132 has no effect in EdU negative cells.
5. The effects on p21 and Cdt1 levels after DCAF14 knockdown are quite minimal in Figure 3B.
6. For Figure 3B and S2A, the authors should also show EdU- cells, to demonstrate that they observe the dramatic downregulation of Cdt1, SET8, and p21 normally observed in S phase.
7. In Figure 3D and E, the drop in Set8 levels after DCAF14 silencing is the same in the presence and absence of Cdt2, arguing that the drop has nothing to do with Cdt2 levels, contrary to the authors' conclusion (same in Figure 3H, after MLN4924 treatment). In contrast, the expected result is actually shown in Figure S2C.
8. The authors repeatedly make claims (e.g. again in 3F) that imply that effects are S phase specific, but never show the non S-phase cells.
9. MLN4924 is not rescuing SET8 levels in S2G.
10. The PLA assay in Figure 5C is missing a key control, omitting EdU.
11. Several of the horizontal black lines on the graphs in Figure S4D (and elsewhere) do not appear to represent the underlying data.
12. In every case where siRNA is used, the authors need to explain how they ruled out off-target effects.

Reviewer #2 Review

Comments to the Authors (Required):

This manuscript shows that loss of the CRL4 substrate adaptor protein DCAF14 results in increased degradation of CRL4-CDT2 substrates, such as SET8, p21, and CDT1. This is most likely due to the competition between CRL4 adaptor proteins, but the mechanism is not explained in this study. Previous studies have shown that loss of DCAF14 increases the degradation of nascent DNA at stalled replication forks. This study went a step further and showed that the effects of DCAF14 on nascent DNA is attributed to an increase of CDT2-mediated protein degradation. Interestingly, this study found that loss of SET8 but not other CDT2 substrates also leads to nascent DNA degradation, which suggests that the effects of DCAF14 loss on stalled forks may be mediated by the reduction in SET8. The authors provide several lines of correlative evidence to support this model.

The most interesting finding of this study is that SET8 may have a role in stabilizing replication forks. However, how DCAF14 affects CDT2-mediated SET8 degradation is not explained. How SET8 stabilizes stalled forks is not explained either. Many important questions are unanswered. For example, is the methyltransferase activity of SET8 required? Is H4K20me1 required? If SET8 is degraded by PCNA-CRL4-CDT2 at replication forks, how can it stabilize the forks? Furthermore, the data in this study do not explain how SET8 responds to replication stress. Overall, this is an interesting study but it is still premature for a strong publication.

1. In Fig. 1C, why was SET8 level reduced by UNC0379? It does not make sense that a substrate competitive inhibitor of SET8 reduces SET8 protein.
2. In Fig. 2D and 2E, EdU- nuclei should be analyzed as controls.
3. In Fig. 2E and 3B, are the differences between siNT and siDCAF14 significant in the presence of MG132 (sample #3 vs #4)? It did not look like MG132 overrides the effects of siDCAF14. The data do not support the authors' interpretation.
4. It is surprising that high HU did not affect CDT2 substrates in Fig. 3A. HU should induce PCNA unloading and reduce the activity of CRL4-CDT2. The authors should check PCNA, CRL4, and CDT2 levels on chromatin in siDCAF14 cells.
5. In Fig. 3E, F, G, and H, EdU- nuclei should be analyzed as controls.
6. The MLN4929 data in Fig. 3H, S2F and S2G are not interpreted correctly. These data only suggest that neddylation is required for the degradation of CDT2 substrates. The data in Fig. S2E only shows that CUL4A neddylation is inhibited by MLN4929. These data do not show that DCAF14 affects neddylation. Because the activity of CRL4 is dependent on neddylation anyway, it is meaningless to suggest DCAF14 affects "neddylated" CRL4-CDT2 .
7. In Fig. 4, it would be necessary to show that the suppression of siDCAF14 effects by siCDT2 is specific. For example,

siBRCA2 should also lead to degradation of nascent DNA and increase of beaks. Can siCDT2 suppress these effects of siBRCA2?

8. Previous studies suggested that loss of BRCA2 increases nascent DNA degradation but does not affect fork restart (Schlacher et al. 2011). Can the authors explain why siDCAF14 affects both nascent DNA degradation and fork restart (Fig. 4E)?

9. In Fig. 6, many of the rescue experiments using MG132 and p97i are not very specific. Can SET8 overexpression alone rescue siDCAF14 in all the assays?

July 6, 2023

Re: Life Science Alliance manuscript #LSA-2023-02230-T

Dr. Huzefa Dungrawala
University of South Florida
MBS
4202 E Fowler Ave
ISA 6202
Tampa, Florida 33620

Dear Dr. Dungrawala,

Thank you for submitting your manuscript entitled "DCAF14 regulates CDT2 to promote SET8-dependent replication fork protection" to Life Science Alliance.

The manuscript was submitted and reviewed at another journal. The authors then chose to transfer their manuscript, along with the reviewers' comments and a proposed revision plan to Life Science Alliance (LSA). The reviewer comments and revision plan was assessed at LSA, and LSA editors deemed that the manuscript could be further considered at LSA provided the authors revise the manuscript, in accordance to what they have laid out in the pbp rebuttal / revision plan.

We, thus, encourage you to submit a revised manuscript to us that includes all the experiments you have laid out in their Revision plan. Given that new data will be added to the revised manuscript, such a revision might need to be re-reviewed, in which case, we will walk the Reviewers through our transfer process.

Thank you for this interesting contribution to Life Science Alliance. We are looking forward to receiving your revised manuscript.

Sincerely,

-- Summary blurb (enter in submission system): A short text summarizing in a single sentence the study (max. 200 characters including spaces). This text is used in conjunction with the titles of papers, hence should be informative and complementary to

the title and running title. It should describe the context and significance of the findings for a general readership; it should be written in the present tense and refer to the work in the third person. Author names should not be mentioned.

B. MANUSCRIPT ORGANIZATION AND FORMATTING:

Reviewer #1 (Comments to the Authors (Required)):

This paper looks at the interplay of two Cul4 E3 ubiquitin ligases, CRL4DCAF14 and CRL4Cdt2, in regulating fork stability. The authors show that fork degradation in HU caused by DCAF14 knock-down is suppressed by also knocking down Cdt2 (or adding a CRL4 inhibitor), implying that CRL4Cdt2 is de-regulated in the absence of CRL4DCAF14. They then also present various lines of evidence that de-regulation of CRL4Cdt2 leads to loss of Set8, a known CRL4Cdt2 substrate in S phase. This leads to a model in which Set8 levels, which are normally low in S phase due to PCNA-coupled degradation by CRL4Cdt2, are lowered even further in the absence of CRL4DCAF14. In other words, even the low levels of Set8 that are normally present are doing something to preserve stalled forks because further reduction of Set8 is not tolerated. They also show evidence that in the absence of CRL4DCAF14, fork degradation involves for reversal and the nucleases DNA2 and MRE11.

This is a potentially interesting manuscript. However, much of the data, especially in the first half of the paper purporting to show Set8 destabilization in the absence of DCAF14, is not convincing, as outlined below. Also, the logic of the final experiments presented in Figure 6 is very convoluted. It would be simpler and more compelling to overexpress Set8 (WT, or a degron mutant), to show that this is sufficient to suppress the fork stability defects seen after DCAF14 loss. Finally, major aspects remain unexplained: how does Set8 protect fork stability? How does DCAF14 suppress CRL4Cdt2 activity?

We thank the reviewer for expressing interest in the manuscript and providing constructive feedback. Admittedly, we do not have detailed mechanisms explaining how SET8 mediates fork protection and how CRL4^{CDT2} activity is regulated by DCAF14. To understand SET8 function, considerable experiments will be required to determine functional domains of SET8 that are necessary for fork protection, how SET8 is transiently stabilized at forks and the enzymatic substrate of SET8 that protects forks from nucleolytic digestion. Our preliminary analysis indicates that SET8 overexpression is not sufficient to restore fork protection in DCAF14-deficient conditions (please see reviewer 2, comment #9). With respect to CDT2 regulation, we carried out additional experiments to determine whether CDT2 is a proteolytic target of DCAF14. As shown below, CDT2 levels are not altered when DCAF14 is transiently depleted across five biological replicates. The following figure is included in the manuscript as Figure 3I. We speculate that CDT2 function is fine-tuned by DCAF14 and testing this model will require targeted proteomic analyses coupled with extensive mutational analyses. Although the questions raised by the reviewer are interesting and critical, the significance of the manuscript is identifying a need to regulate CDT2 function at stalled forks to allow SET8 to mediate fork protection. These are the most important results of the manuscript and provide a critical conceptual advance.

Specific Points

1. Figure 1E is not convincing because there was clearly a Western transfer problem in the key, middle lane in the H4K20me1 blot. Also, H4 looks overloaded in lanes 3 and 4. Please redo and show lighter exposure of loading controls next time.

We have performed H4K20me1 immunoblotting experiments by DCAF14 complementation in cells transfected with siRNAs targeting 5'UTR region of DCAF14. As shown below, loss of DCAF14 reduces H4K20me1 levels that are partially restored by re-expressing DCAF14. The following figure is included in the manuscript as Figure 1E.

2. The lowered expression of SET8 in Figure 2A is not convincing because Ku70, which is presumably a loading control, is also lower in the siDCAF14 lanes. Hard to judge how much because again, blot is overexposed. SET8 levels need to be normalized to the loading control, but not when it's overexposed as in this figure. In Figure S1B, where Ku loading is equal, there is barely any reduction in SET8.

We thank the reviewer for raising this concern. Although the probing is taken in higher exposure, it is within the dynamic range of detection by LICOR imaging system. Included below are quantifications from multiple western blots comparing siINT and siDCAF14 U2OS cells for all CDT2 substrates SET8 (n=4), p21 (n=4) and CDT1 (n=4). The data is consistent with the observations that loss of DCAF14 reduces levels of CDT2 substrates. The figure panels 2B and 3B are now added to the manuscript.

3. There is a dramatic drop in G2 population in U2OS cells in Figure S1A. Why is this?

U2OS and siNT populations in Figure S1A exhibit nearly identical G2 percentages of 9.1% and 10% respectively. We also identified that G2% for DCAF14 KO was incorrectly assigned 1.1 instead of 11.1. The number has been amended in Figure S1A.

4. To make the claim that the rescue of SET8 levels by MG132 is "coupled to DNA synthesis" they would need to show that MG132 has no effect in EdU negative cells.

We thank the reviewer for raising this point. As suggested, we have performed experiments with MG132 and MLN4924 (please see reviewer 2, comment #5) to determine whether these compounds have effects outside of S-phase. As expected, endogenous SET8 level is lower in replicating cells (left panel). We also find that acute, 2 hour exposure to MG132 (or MLN4924) in control populations increase SET8 levels in both replicating and non-replicating cells (right panel) consistent with previous observations showing SET8 degradation is CRL4^{CDT2} dependent in S-phase (Centore et al., *Mol Cell* 2010, Abbas et al., *Mol Cell* 2010), APC^{CDH1} dependent in mitosis (Wu et al., *Genes Dev.* 2010) and SCF^{B-TRCP} dependent in G1 phase (Wang et al., *Nat Commun* 2015). Our data also indicate that MG132 restores SET8 levels in siDCAF14-deficient cells in both EdU+ and EdU- nuclei. These panels are added to the revised manuscript as Figure 2E and 2F. We have removed the conclusion that the rescue of SET8 levels is exclusive to DNA synthesis and instead include the following: *DCAF14 absence causes increased proteasomal turnover of SET8 during DNA replication and outside of S-phase.*

5. The effects on p21 and Cdt1 levels after DCAF14 knockdown are quite minimal in Figure 3B.

The effects of DCAF14 knockdown on levels of CDT2 substrates are highly reproducible over several biological repeats as shown in Figures 3D, 3G and S2F. The decrease in levels of CDT2 substrates is also observable in DCAF14 knockout cells as shown in Figure 3E. We also quantified intensities from two additional biological repeats for CDT1 (green) and one additional repeat for p21 (blue) shown below.

6. For Figure 3B and S2A, the authors should also show EdU- cells, to demonstrate that they observe the dramatic downregulation of Cdt1, SET8, and p21 normally observed in S phase.

As requested by the reviewer, quantitative imaging of CDT2 substrates in EdU- cells is included in Figure 2E for SET8 (refer comment #4) and Figure 3C below for CDT1 and p21.

7. In Figure 3D and E, the drop in Set8 levels after DCAF14 silencing is the same in the presence and absence of Cdt2, arguing that the drop has nothing to do with Cdt2 levels, contrary to the authors' conclusion (same in Figure 3H, after MLN4924 treatment). In contrast, the expected result is actually shown in Figure S2C.

We respectfully disagree. SET8 levels in DCAF14-/CDT2- conditions are higher than when DCAF14 is depleted alone (Figure 3G). Thus, the higher drop in SET8 levels for DCAF14-/CDT2+ conditions can be explained by an increase in CDT2 activity. In agreement with this model, all substrate levels (SET8, p21 and CDT1) in DCAF14-deficient cells can be restored when CDT2 is co-depleted or acutely inhibited by MLN4924 or MG132. Results in figure S2C are also consistent with this model as SET8 levels in DCAF14-deficient cells are higher when CDT2 is depleted. The extent in SET8 rescue in presence of HU is higher

compared to untreated conditions and perhaps reflective of increased SET8 turnover that is dependent on CDT2.

8. The authors repeatedly make claims (e.g. again in 3F) that imply that effects are S phase specific, but never show the non S-phase cells.

We agree with the reviewer comments regarding this figure. The quantitative imaging data depicting increase in SET8 levels upon DCAF14 overexpression has been revised to show SET8 intensities in all DAPI stained nuclei. Our experiments involve DCAF14 overexpression under a constitutive promoter thereby restricting our ability to assess cell cycle specific changes in SET8 levels when DCAF14 is overexpressed. The conclusion has been accordingly revised and the following figure has been added to the manuscript (Figure 3H). We also apologize for conveying the conclusion that the effects are S-phase specific. Rather, we infer that the effects occur in both replicating and non-replicating cells and that the S-phase related phenotypes are due to increased CDT2 activity.

9. MLN4924 is not rescuing SET8 levels in S2G.

We thank the reviewer for raising this point. We repeated the experiments and observe that MLN4924 exposure rescues SET8 levels in HU treated cells. The following representative graph is added in S2G in the manuscript.

10. The PLA assay in Figure 5C is missing a key control, omitting EdU.

Shown below is the PLA assay to test proximity of SET8 to Biotin, which includes a EdU- control as requested by the reviewer. The following figure is now added to the revised manuscript as Figure 5C.

11. Several of the horizontal black lines on the graphs in Figure S4D (and elsewhere) do not appear to represent the underlying data.

All figure legends, including quantification and statistical analyses, describe the statistical measures utilized to plot the graphs. We have revisited the Graphpad prism files and confirmed the represented 'mean ± SEM' values are accurate.

12. In every case where siRNA is used, the authors need to explain how they ruled out off-target effects.

siRNAs targeting DCAF14 used in this study have been previously validated to be highly efficient in downregulating DCAF14 and exhibit similar phenotypic effects to DCAF14 knockout clones (Townsend et al., *Cell Reports* 2021). We have validated phenotypes associated with SET8 depletion using two independent siRNAs and chemical inhibitor UNC0379 in this study. siRNAs targeting MRE11, DNA2, ZRANB3, SMARCAL1, WDR5 and RBBP5 are extensively utilized in the field and have been shown to have minimal off-target effects.

We performed CDT1 and p21 depletions with alternate siRNAs as suggested. In each case, the depletions are comparable and exhibit similar phenotypes (top panel, Figures S4B and S4C). We also show that depleting CDT2 with an alternate siRNA rescues nascent strand degradation in siDCAF14-transfected cells (bottom panel, Figures S3E and S3F). These data are now included in the revised manuscript.

Reviewer #2 (Comments to the Authors (Required)):

This manuscript shows that loss of the CRL4 substrate adaptor protein DCAF14 results in increased degradation of CRL4-CDT2 substrates, such as SET8, p21, and CDT1. This is most likely due to the competition between CRL4 adaptor proteins, but the mechanism is not explained in this study. Previous studies have shown that loss of DCAF14 increases the degradation of nascent DNA at stalled replication forks. This study went a step further and showed that the effects of DCAF14 on nascent DNA is attributed to an increase of CDT2-mediated protein degradation. Interestingly, this study found that loss of SET8 but not other CDT2 substrates also leads to nascent DNA degradation, which suggests that the effects of DCAF14 loss on stalled forks may be mediated by the reduction in SET8. The authors provide several lines of correlative evidence to support this model.

The most interesting finding of this study is that SET8 may have a role in stabilizing replication forks. However, how DCAF14 affects CDT2-mediated SET8 degradation is not explained. How SET8 stabilizes stalled forks is not explained either. Many important questions are unanswered. For example, is the methyltransferase activity of SET8 is required? Is H4K20me1 required? If SET8 is degraded by PCNA-

CRL4-CDT2 at replication forks, how can it stabilize the forks? Furthermore, the data in this study do not explain how SET8 responds to replication stress. Overall, this is an interesting study but it is still premature for a strong publication.

We thank the reviewer for finding the study interesting and providing constructive feedback. The questions pointed out by this reviewer concur with reviewer 1 as well and are very relevant to understanding how DCAF14 regulates CDT2 to help SET8 mediate replication fork stability. We are actively pursuing mechanistic analyses to determine how CRL4^{CDT2} function is modulated and the functional contribution of SET8 in stalled fork protection. Functional analyses of SET8 in replication stress conditions will require an inducible system to test the dependencies on the catalytic function and PCNA interaction. We are also designing a degron system to acutely degrade DCAF14 in order to elucidate its role in S-phase.

1. In Fig. 1C, why was SET8 level reduced by UNC0379? It does not make sense that a substrate competitive inhibitor of SET8 reduces SET8 protein.

We currently lack mechanistic insight into why acute treatment of cells with UNC0379 downregulates SET8 protein levels. However, our observations are consistent with other studies that have reported similar effects on SET8 with UNC0379 (Wu et. al., *Scientific Reports* 2020, Veschi et. al., *Cancer Cell* 2017). Further studies will be needed to delineate the cause of the acute SET8 downregulation upon addition of UNC0379.

2. In Fig. 2D and 2E, EdU- nuclei should be analyzed as controls.

EdU- nuclei were analyzed as controls and the following figure is now added to the manuscript as Figure 2E and 2F.

3. In Fig. 2E and 3B, are the differences between siNT and siDCAF14 significant in the presence of MG132 (sample #3 vs #4)? It did not look like MG132 overrides the effects of siDCAF14. The data do not support the authors' interpretation.

We thank the reviewer for raising this point. We performed the statistical analyses between siNT and siDCAF14 in the presence of MG132 for all substrates as requested and added the revised figures. The MG132-dependent rescue in substrate levels and fork degradation in siDCAF14 conditions is consistent

with results observed in experiments using p97i (Figures 6D and 6E) indicating that enhanced proteasomal degradation of CDT2 substrates occurs when DCAF14 is lost. Since the IF experiments are performed using asynchronous populations, the substrate levels are inherently lower in DCAF14-depleted cells prior to addition of MG132 and thereby not fully restored since the substrates are completely digested. This likely explains why complete restoration of substrate levels cannot be observed in MG132-treated, DCAF14-deficient cells.

To take these observations into account, we have modified our results to “...proteasomal inhibitor MG132 largely restores CDT2 substrate levels in DCAF14-silenced cells that are left unchallenged.” The ability of p97i to fully restore substrate levels in Figure 6C further supports the conclusion that loss of CDT2 substrates in siDCAF14 is mostly driven by proteasome.

4. It is surprising that high HU did not affect CDT2 substrates in Fig. 3A. HU should induce PCNA unloading and reduce the activity of CRL4-CDT2. The authors should check PCNA, CRL4, and CDT2 levels on chromatin in siDCAF14 cells.

The initial model we tested was that DCAF14 serves as a substitute for CDT2 at stalled forks since CDT2 activity is expected to reduce due to PCNA unloading. Contrary to the prediction that depleting DCAF14 should stabilize CDT2 substrates, we find that CDT2 substrate levels are instead further downregulated in the absence of DCAF14. Although PCNA is actively unloaded in response to high-dose HU, PCNA continues to associate with HU stalled forks to participate in translesion synthesis, fork reversal and fork protection. Presumably the PCNA bound to DNA is sufficient to trigger proteolytic destruction. Indeed, CDT2 continues to degrade substrates even in the presence of replication stress (Centore et al., *Mol Cell* 2010, Nishitani et al., *J Biol Chem* 2008) but how the activity is modulated despite PCNA unloading is unknown. Consistent with these observations, we find that CDT2 substrate intensities in HU-stressed nuclei are mostly unaltered except for p21 which occurs due to stress-induced increased in transcription (see figure below). These changes are unlike the elevated substrate levels seen when CDT2 is depleted (Figure S2C) further indicating that CDT2 function is not substantially reduced when forks are stalled. Continued CDT2-dependent proteolytic destruction is thus vital to prevent genome instability due to re-replication.

We are currently pursuing iPOND-SILAC mass spectrometry analyses to study the fork proteome alterations in DCAF14 knockout cells.

[Figure removed by editorial staff per authors' request]

5. In Fig. 3E, F, G, and H, EdU- nuclei should be analyzed as controls.

Figure 3F is revised to show levels of mean SET8 intensity in DAPI nuclei (please see reviewer 1, comment #8). Analysis of EdU- nuclei was performed and the following figures were added to the manuscript for experiments with MLN4924 (Figure S1D) and PLA analyses for PCNA-CDT2 proximity (Figure 3J).

6. The MLN4929 data in Fig. 3H, S2F and S2G are not interpreted correctly. These data only suggest that neddylation is required for the degradation of CDT2 substrates. The data in Fig. S2E only shows that CUL4A neddylation is inhibited by MLN4929. These data do not show that DCAF14 affects neddylation. Because the activity of CRL4 is dependent on neddylation anyway, it is meaningless to suggest DCAF14 affects "neddylated" CRL4-CDT2 .

We agree with the reviewer and removed the term neddylated to revise the conclusion as follows: “in cells without DCAF14, increased activity of CRL4^{CDT2} appears to be responsible for elevated loss of CDT2 substrates.”

7. In Fig. 4, it would be necessary to show that the suppression of siDCAF14 effects by siCDT2 is specific. For example, siBRCA2 should also lead to degradation of nascent DNA and increase of beaks. Can siCDT2 suppress these effects of siBRCA2?

We have performed the experiment suggested by the reviewer. Loss of CDT2 does not restore fork protection in BRCA2-depleted cells indicating that CDT2 triggers nascent strand degradation specifically in conditions of DCAF14 deficiency. The following figure is added to the revised manuscript (Figures 6B and S5B).

8. Previous studies suggested that loss of BRCA2 increases nascent DNA degradation but does not affect fork restart (Schlachter et al. 2011). Can the authors explain why siDCAF14 affects both nascent DNA degradation and fork restart (Fig. 4E)?

DCAF14-deficient cells exhibit decreased stalled fork restart due to irreversible collapse into double strand breaks resulting in increased sensitivity to replication stress reagents (Townsend et. al., *Cell Reports* 2021). Since nascent strand degradation in DCAF14-silenced cells is fork reversal dependent, digestion of nascent DNA presumably exposes remodeled forks to nuclease-dependent DSB formation thereby inactivating replication forks.

9. In Fig. 6, many of the rescue experiments using MG132 and p97i are not very specific. Can SET8 overexpression alone rescue siDCAF14 in all the assays?

As requested by both reviewers, we have performed fork protection assays in conditions of SET8 overexpression. While SET8 overexpression marginally affects fork protection in control populations, SET8 overexpression is not sufficient to rescue nascent strand degradation in DCAF14-deficient cells. Interestingly, SET8 levels are also reduced when overexpressed in cells without DCAF14 supporting our observations that DCAF14 regulates SET8 stability. To establish whether this phenomenon is dependent on CDT2 activity, and whether it is observable with other substrates, will require further detailed investigation. Considering these results are preliminary, we prefer not to include this data in our

manuscript. Constitutive expression of SET8 also interferes with DNA replication and causes replication stress (Centore et al., *Mol Cell* 2010). To overcome these limitations, we are working on generating cell lines with inducible SET8 (WT, PIP Δ and CD) expression that will help determine the contribution of SET8 in replication fork protection.

[Figure removed by editorial staff per authors' request].

October 17, 2023

RE: Life Science Alliance Manuscript #LSA-2023-02230-TR

Dr. Huzefa Dungrawala
University of South Florida
MBS
4202 E Fowler Ave
ISA 6202
Tampa, Florida 33620

Dear Dr. Dungrawala,

Thank you for submitting your revised manuscript entitled "DCAF14 regulates CDT2 to promote SET8-dependent replication fork protection". We would be happy to publish your paper in Life Science Alliance pending final revisions necessary to meet our formatting guidelines.

- please add the Twitter handle of your host institute/organization as well as your own or/and one of the authors in our system
- please use the [10 author names et al.] format in your references (i.e., limit the author names to the first 10)
- please add a callout for Figure S4D to your main manuscript text

Figure Checks:

- please add scale bars to Figures 2D and 4E
- please provide the source data for Figure S3E

A. FINAL FILES:

B. MANUSCRIPT ORGANIZATION AND FORMATTING:

Sincerely,

October 26, 2023

RE: Life Science Alliance Manuscript #LSA-2023-02230-TRR

Dr. Huzefa Dungrawala
University of South Florida
MBS
4202 E Fowler Ave
ISA 6202
Tampa, Florida 33620

Dear Dr. Dungrawala,

Thank you for submitting your Research Article entitled "DCAF14 regulates CDT2 to promote SET8-dependent replication fork protection". It is a pleasure to let you know that your manuscript is now accepted for publication in Life Science Alliance. Congratulations on this interesting work.

DISTRIBUTION OF MATERIALS:

Again, congratulations on a very nice paper. I hope you found the review process to be constructive and are pleased with how the manuscript was handled editorially. We look forward to future exciting submissions from your lab.

Sincerely,
